# Towards Self-Interpretable
# Graph-Level Anomaly Detection

**Yixin Liu[1], Kaize Ding[2], Qinghua Lu[3], Fuyi Li[4,5], Leo Yu Zhang[6], Shirui Pan[6]***
[1]Monash University, [2]Northwestern University, [3]Data61, CSIRO,
[4]Northwest A&F University, [5]The University of Adelaide, [6]Griffith University
`yixin.liu@monash.edu, kaize.ding@northwestern.edu, qinghua.lu@data61.csiro.au,`
`fuyi.li@nwsuaf.edu.cn, leo.zhang@griffith.edu.au, s.pan@griffth.edu.au`

## Abstract

Graph-level anomaly detection (GLAD) aims to identify graphs that exhibit notable dissimilarity compared to the majority in a collection. However, current works primarily focus on evaluating graph-level abnormality while failing to provide meaningful explanations for the predictions, which largely limits their reliability and application scope. In this paper, we investigate a new challenging problem, *explainable GLAD*, where the learning objective is to predict the abnormality of each graph sample with corresponding explanations, i.e., the vital subgraph that leads to the predictions. To address this challenging problem, we propose a Self-Interpretable Graph aNomaly dETection model (`SIGNET` for short) that detects anomalous graphs as well as generates informative explanations simultaneously. Specifically, we first introduce the multi-view subgraph information bottleneck (MSIB) framework, serving as the design basis of our self-interpretable GLAD approach. This way `SIGNET` is able to not only measure the abnormality of each graph based on cross-view mutual information but also provide informative graph rationales by extracting bottleneck subgraphs from the input graph and its dual hypergraph in a self-supervised way. Extensive experiments on 16 datasets demonstrate the anomaly detection capability and self-interpretability of `SIGNET`.

## 1 Introduction

Graphs are ubiquitous data structures in numerous domains, including chemistry, traffic, and social networks [1, 2, 3]. Among machine learning tasks for graph data, graph-level anomaly detection (GLAD) is a challenge that aims to identify the graphs that exhibit substantial dissimilarity from the majority of graphs in a collection [4]. GLAD presents great potential for various real-world scenarios, such as toxic molecule recognition [5] and pathogenic brain mechanism discovery [6]. Recently, GLAD has drawn increasing research attention, with advanced techniques being applied to this task, e.g., knowledge distillation [4] and one-class classification [7].

Despite their promising performance, existing works [4, 7, 8, 9] mainly aim to answer **how** to predict abnormal graphs by designing various GLAD architectures; however, they fail to provide explanations for the prediction, i.e., illustrating **why** these graphs are recognized as anomalies. In real-world applications, it is of great significance to make anomaly detection models explainable [10]. From the perspective of models, valid explainability makes GLAD models trustworthy to meet safety and security requirements [11]. For example, an explainable fraud detection model can pinpoint specific fraudulent behaviors when identifying defrauders, which enhances the reliability of predictions. From the perspective of data, an anomaly detection model with explainability can

---

*Corresponding Author.

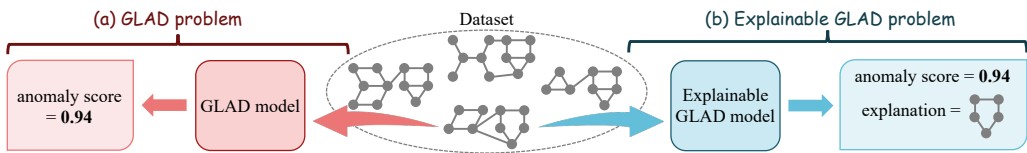

Figure 1: A toy example to illustrate (a) GLAD problem and (b) explainable GLAD problem.

help us explicitly understand the anomalous patterns of the dataset, which further supports human experts in data understanding [12]. For instance, an explainable GLAD model for molecules can summarize the functional groups that cause abnormality, enabling researchers to deeply investigate the properties of compounds. Hence, the broad applications of interpreting anomaly detection results motivate us to investigate the problem of **Explainable GLAD** where the GLAD model is expected to measure the abnormality of each graph sample as well as provide meaningful explanations of the predictions during the inference time. As an example shown in Fig. 1, the GLAD model also extracts a graph rationale [13, 14] corresponding to the predicted anomaly score. Although there are a few studies [10, 15, 16] proposed to explain anomaly detection results for visual or tabular data, explainable GLAD remains underexplored and it is non-trivial to apply those methods to our problem due to the discrete nature of irregular graph-structured data [17].

Towards the goal of designing an explainable GLAD model, two essential challenges need to be solved with careful design: *Challenge 1 — how to make the GLAD model self-interpretable*[2]*?* Even though we can leverage existing post-hoc explainers [19, 20] for GNNs to explain the predictions of the GLAD model, such post-hoc explainers are not synergistically learned with the detection models, resulting in the risk of wrong, biased, and sub-optimal explanations [21, 22]. Hence, developing a self-interpretable GLAD model which detects graph-level anomalies with explanations inherently is more desirable and requires urgent research efforts. *Challenge 2 — how to learn meaningful graph explanations without using supervision signals?* For the problem of GLAD, ground-truth anomalies are usually unavailable during training, raising significant challenges to both detecting anomalies and providing meaningful explanations. Since most of the existing self-interpretable GNNs [13, 21, 22] merely focus on the (semi-)supervised setting, in particular the node/graph classification tasks, how to design a self-interpretable model for the explainable GLAD problem where ground-truth labels are inaccessible remains a challenging task.

To solve the above challenges, in this paper, we develop a novel Self-Interpretable Graph aNomaly dETection model (`SIGNET` for short). Based on the information bottleneck (IB) principle, we first propose a multi-view subgraph information bottleneck (MSIB) framework, serving as the design basis of our self-interpretable GLAD model. Under the MSIB framework, the instantiated GLAD model is able to predict the abnormality of each graph as well as generate corresponding explanations without relying on ground-truth anomalies simultaneously. To learn the self-interpretable GLAD model without ground-truth anomalies, we introduce the dual hypergraph as a supplemental view of the original graph and employ a unified bottleneck subgraph extractor to extract corresponding graph rationales. By further conducting multi-view learning among the extracted graph rationales, `SIGNET` is able to learn the feature patterns from both node and edge perspectives in a purely self-supervised manner. During the test phase, we can directly measure the abnormality of each graph sample based on its inter-view agreement (i.e., cross-view mutual information) and derive the corresponding graph rationales for the purpose of explaining the prediction. To sum up, our contribution is three-fold:

- **Problem.** We propose to investigate the explainable GLAD problem that has broad application prospects. To the best of our knowledge, this is the *first* attempt to study the explainability problem for graph-level anomaly detection.
- **Algorithm.** We propose a novel self-interpretable GLAD model termed `SIGNET`, which infers graph-level anomaly scores and subgraph-level explanations simultaneously with the multi-view subgraph information bottleneck framework.

---

[2]In this paper, we distinguish the terms "explainability" and "interpretability" following a recent survey paper [18]: "explainable artificial intelligence" is a widespread and high-level concept, hence we define the research problem as "explainable GLAD"; for the model that can provide interpretations of the predictions of itself, we consider it as "interpretable" or "self-interpretable". Detailed definitions are provided in Appendix A.

- **Evaluation.** We perform extensive experiments to corroborate the anomaly detection performance and self-interpretation ability of `SIGNET` via thorough comparisons with state-of-the-art methods on 16 benchmark datasets.

## 2 Preliminaries and Related Work

In this section, we introduce the preliminaries and briefly review the related works. A more comprehensive literature review can be found in Appendix B.

**Notations.** Let $G = (\mathcal{V}, \mathcal{E}, \mathbf{X})$ be a simple graph with $n$ nodes and $m$ edges, where $\mathcal{V}$ is the set of nodes and $\mathcal{E}$ is the set of edges. The node features are included by feature matrix $\mathbf{X} \in \mathbb{R}^{n \times d_f}$, and the connectivity among the nodes is represented by adjacency matrix $\mathbf{A} \in \mathbb{R}^{n \times n}$. Unlike simple graphs where each edge only connects two nodes, "hypergraph" is a generalization of a traditional graph structure in which hyperedges connect more than two nodes. We define a hypergraph with $n^*$ nodes and $m^*$ hyperedges as $G^* = (\mathcal{V}^*, \mathcal{E}^*, \mathbf{X}^*)$, where $\mathcal{V}^*$, $\mathcal{E}^*$, and $\mathbf{X}^* \in \mathbb{R}^{n^* \times d_f^*}$ are the node set, hyperedge set, and node feature matrix respectively. To indicate the higher-order relations among arbitrary numbers of nodes within a hypergraph, we use an incidence matrix $\mathbf{M}^* \in \mathbb{R}^{n^* \times m^*}$ to represent the interaction between $n^*$ nodes and $m^*$ hyperedges. Alternatively, a simple graph and a hypergraph and be represented by $G = (\mathbf{A}, \mathbf{X})$ and $G^* = (\mathbf{M}^*, \mathbf{X}^*)$, respectively. We denote the Shannon mutual information (MI) of two random variables $A$ and $B$ as $I(A; B)$.

**Graph Neural Networks (GNNs).** GNNs are the extension of deep neural networks onto graph data, which have been applied to various graph learning tasks [1, 2, 23, 24, 25, 26, 27, 28]. Mainstream GNNs usually follow the paradigm of message passing [2, 23, 24, 26]. Some studies termed hypergraph neural networks (HGNNs) also apply GNNs to hypergraphs [29, 30, 31]. The formulations of GNN and HGNN are in Appendix C. To make the predictions understandable, some efforts try to uncover the explanation for GNNs [18, 32]. A branch of methods, termed post-hoc GNN explainers, use specialized models to explain the behavior of a trained GNN [19, 20, 33]. Meanwhile, some self-interpretable GNNs can intrinsically provide explanations for predictions using interpretable designs in GNN architectures [13, 21, 22]. While these methods mainly aim at supervised classification scenarios, how to interpret unsupervised anomaly detection models still remains open.

**Information Bottleneck (IB).** IB is an information theory-based approach for representation learning that trains the encoder by preserving the information that is relevant to label prediction while minimizing the amount of superfluous information [34, 35, 36]. Formally, given the data $X$ and the label $Y$, IB principle aims to find the representation $Z$ by maximizing the following objective: $\max_Z I(Z; Y) - \beta I(X; Z)$, where $\beta$ is a hyper-parameter to trade off informativeness and compression. To extend IB onto unsupervised learning scenarios, Multi-view Information Bottleneck (MIB) [37] provides an optimizable target for unsupervised multi-view learning, which alleviates the reliance on label $Y$. Given two different and distinguishable views $V_1$ and $V_2$ of the same data $X$, the objective of MIB is to learn sufficient and compact representations $Z_1$ and $Z_2$ for two views respectively. Taking view $V_1$ as an example, by factorizing the MI between $V_1$ and $Z_1$, we can identify two components: $I(V_1; Z_1) = I(V_1; Z_1|V_2) + I(V_2; Z_1)$, where the first term is the superfluous information that is expected to be minimized, and the second term is the predictive information that should be maximized. Then, $Z_1$ can be learned using a relaxed Lagrangian objective:

$$\max_{Z_1} I(V_2; Z_1) - \beta_1 I(V_1; Z_1|V_2), \tag{1}$$

where $\beta_1$ is a trade-off parameter. By optimizing Eq. (1) and its counterpart in view $V_2$, we can learn informative and compact $Z_1$ and $Z_2$ by extracting the information from each other.

IB principle is also proven to be effective in graph learning tasks, such as graph contrastive learning [38, 39], subgraph recognition [17, 40], graph-based recommendation [41], and robust graph representation learning [22, 42, 43]. Nevertheless, how to leverage the idea of IB on graph anomaly detection tasks is still an open problem.

**Graph-level Anomaly Detection (GLAD).** GLAD aims to recognize anomalous graphs from a set of graphs by learning an anomaly score for each graph sample to indicate its degree of abnormality [4, 7, 8, 9]. Recent studies try to address the GLAD problem with various advanced techniques, such as knowledge distillation [4], one-class classification [7], transformation learning [8],

and deep graph kernel [44]. However, these methods can only learn the anomaly score but fail to provide explanations, i.e., the graph rationale causing the abnormality, for their predictions.

**Problem Formulation.** Based on this mainstream unsupervised GLAD paradigm [4, 7, 8], in this paper, we present a novel research problem termed *explainable GLAD*, where the GLAD model is expected to provide the anomaly score as well as the explanations of such a prediction for each testing graph sample. Formally, the proposed research problem can be formulated by:

**Definition 2.1** (Explainable graph-level anomaly detection). Given the training set $\mathcal{G}_{tr}$ that contains a number of normal graphs, we aim at learning an explainable GLAD model $f : \mathbb{G} \to (\mathbb{R}, \mathbb{G})$ that is able to predict the abnormality of a graph and provide corresponding explanations. In specific, given a graph $G_i$ from the test set $\mathcal{G}_{te}$ with normal and abnormal graphs, the model can generate an output pair $f(G_i) = (s_i, G_i^{(es)})$, where $s_i$ is the anomaly score that indicates the abnormality degree of $G_i$, and $G_i^{(es)}$ is the subgraph of $G_i$ that explains why $G_i$ is identified as a normal/abnormal sample.

## 3 Methodology

This section details the proposed model SIGNET for explainable GLAD. Firstly, we derive a multi-view subgraph information bottleneck (MSIB) framework (Sec. 3.1) that allows us to identify anomalies with causal interpretations provided. Then, we provide the instantiation of the components in MSIB framework, including view construction (Sec. 3.2), bottle subgraph extraction (Sec. 3.3), and cross-view mutual information (MI) maximization (Sec. 3.4), which compose the SIGNET model. Finally, we introduce the self-interpretable GLAD inference (Sec. 3.5) of SIGNET. The overall learning pipeline of SIGNET is demonstrated in Fig. 2(a).

### 3.1 MSIB Framework Overview

To achieve the goal of self-interpretable GLAD, an unsupervised learning approach that can jointly predict the abnormality of graphs and yield corresponding explanations is required. Inspired by the concept of information bottleneck (IB) and graph multi-view learning [36, 37, 45], we propose multi-view subgraph information bottleneck (MSIB), a self-interpretable and self-supervised learning framework for GLAD. The learning objective of MSIB is to optimize the "bottleneck subgraphs", the vital substructure on two distinct views of a graph, by maximizing the predictive structural information shared by both graph views while minimizing the superfluous information that is irrelevant to the cross-view agreement. Such an objective can be optimized in a self-supervised manner, without the guidance of ground-truth labels. Due to the observation that latent anomaly patterns of graphs can be effectively captured by multi-view learning [8, 9], we can directly use the cross-view agreement, i.e., the MI between two views, to evaluate the abnormality of a graph sample. Simultaneously, the extracted bottleneck subgraphs provide us with graph rationales to explain the anomaly detection predictions, since they contain the most compact substructure sourced from the original data and the most discriminative knowledge for the predicted abnormality, i.e., the estimated cross-view MI.

Formally, in the proposed MSIB framework, we assume each graph sample $G$ has two different and distinguishable views $G^1$ and $G^2$. Then, taking view $G^1$ as an example, the target of MSIB is to learn a bottleneck subgraph $G^{1(s)}$ for $G^1$ by optimizing the following objective:

$$\max_{G^{1(s)}} I(G^2; G^{1(s)}) - \beta_1 I(G^1; G^{1(s)}|G^2). \tag{2}$$

Similar to MIB (Eq. (1)), the optimization for $G^{2(s)}$, the bottleneck subgraph of $G^2$, can be written in the same format. Then, by parameterizing bottleneck subgraph extraction and unifying the domain of bottleneck subgraphs, the objective can be transferred to minimize a tractable loss function:

$$\mathcal{L}_{MSIB} = -I(G^{1(s)}; G^{2(s)}) + \beta D_{SKL}\left(p_\theta(G^{1(s)}|G^1)\|p_\psi(G^{2(s)}|G^2)\right), \tag{3}$$

where $p_\theta(G^{1(s)}|G^1)$ and $p_\psi(G^{2(s)}|G^2)$ are the bottleneck subgraph extractors (parameterized by $\theta$ and $\psi$) for $G^1$ and $G^2$ respectively, $D_{SKL}(\cdot)$ is the symmetrized Kullback–Leibler (SKL) divergence, and $\beta$ is a trade-off hyper-parameter. Detailed deductions from Eq. (2) to Eq. (3) are in Appendix D.

MSIB framework can guide us to build a self-interpretable GLAD model. The first term in Eq. (3) tries to maximize the MI between the bottleneck subgraphs from two views, which not only prompts the

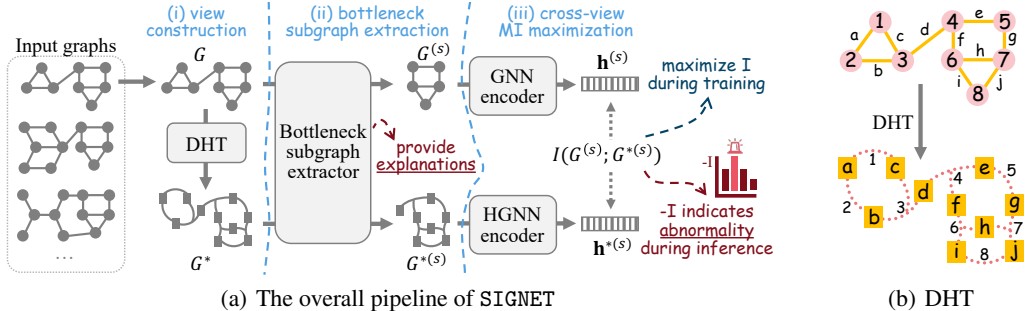

(a) The overall pipeline of SIGNET        (b) DHT

Figure 2: (a) The overall pipeline of the proposed model SIGNET, consisting of (i) view construction, (ii) bottleneck subgraph extraction, and (iii) cross-view MI maximization. (b) An illustration of dual hypergraph transformation (DHT), where the nodes (●) and edges (—) in the original graph correspond to hyperedges (⋯) and nodes (■) in its dual hypergraph, respectively.

model to capture vital subgraphs but also helps capture the cross-view matching patterns for anomaly detection. The second term in Eq. (3) is a regularization term to align the extractors, which ensures the compactness of bottleneck subgraphs. During inference, $-I(G^{1(s)}; G^{2(s)})$ can be regarded as a measure of abnormality. Meanwhile, the bottleneck subgraphs extracted by $p_\theta$ and $p_\psi$ can serve as explanations. In the following subsections, we take SIGNET as a practical implementation of MSIB framework. We illustrate the instantiations of view construction ($G^1$ and $G^2$), subgraph extractors ($p_\theta$ and $p_\psi$), MI estimation ($I(G^{1(s)}; G^{2(s)})$), and explainable GLAD inference, respectively.

## 3.2 Dual Hypergraph-based View Construction

To implement MSIB framework for GLAD, the first step is to construct two different and distinguishable views $G^1$ and $G^2$ for each sample $G$. In multi-view learning approaches [37, 46, 47, 48], a general strategy is using stochastic perturbation-based data augmentation (e.g., edge modification [46] and feature perturbation [47]) to create multiple views. Despite their success in graph representation learning [46, 47, 48], we claim that perturbation-based view constructions are not appropriate in SIGNET for the following reasons. 1) Low sensitivity to anomalies. Due to the similarity of normal and anomalous graphs in real-world data, perturbations may create anomaly-like data from normal data as the augmented view [49]. In this case, maximizing the cross-view MI would result in reduced sensitivity of the model towards distinguishing between normal and abnormal data, hindering the performance of anomaly detection [50]. 2) Less differentiation. In the principle of MIB, two views should be distinguishable and mutually redundant [37]. However, the views created by graph perturbation from the same sample can be similar to each other, which violates the assumption of our basic framework. 3) Harmful instability. The MI $I(G^{1(s)}; G^{2(s)})$ for abnormality measurement is highly related to the contents of two views. Nevertheless, the view contents generated by stochastic perturbation can be quite unstable due to the randomness, leading to inaccurate estimation of abnormality.

Considering the above limitations, a perturbation-free, distinct, and stable strategy is required for view construction. To this end, we utilize *dual hypergraph transformation* (DHT) [51] to construct the opposite view of the original graph. Concretely, for a graph sample $G$, we define the first view as itself (i.e., $G^1 = G$) and the second view as its dual hypergraph $G^*$ (i.e., $G^2 = G^*$). Based on the hypergraph duality [52, 53], dual hypergraph can be acquired from the original simple graph with DHT: each edge of the original graph is transformed into a node of the dual hypergraph, and each node of the original graph is transformed into a hyperedge of the dual hypergraph [51]. As the example shown in Fig. 2(b), the structural roles of nodes and edges are interchanged by DHT, and the incidence matrix $\mathbf{M}^*$ of the dual hypergraph is the transpose of the incidence matrix of the original graph. To initialize the node features $\mathbf{X}^* \in \mathbb{R}^{m \times d_f^*}$ of $G^*$, we can either use the original edge features (if available), or construct edge-level features from the original node features or according to edge-level geometric property.

The DHT-based view construction brings several advantages. Firstly, the dual hypergraph has significantly distinct contents from the original view, which caters to the needs for differentiation in

MIB. Secondly, the dual hypergraph pays more attention to the edge-level information, encouraging the model to capture not only node-level but also edge-level anomaly patterns. Thirdly, DHT is a bijective mapping between two views, avoiding confusion between normal and abnormal samples. Fourthly, DHT is randomness-free, ensuring the stable estimation of MI.

### 3.3 Bottleneck Subgraph Extraction

In MSIB framework, bottleneck subgraph extraction is a key component that learns to refine the core rationale for abnormality explanations. Following the procedure of MSIB, we need to establish two bottleneck subgraph extractors for the original view $G$ and dual hypergraph view $G^*$ respectively. To model the discrete subgraph extraction process in a differentiable manner with neural networks, following previous methods [14, 17, 22], we introduce continuous relaxation into the subgraph extractors. Specifically, for the original view $G$, we model the subgraph extractor with $p_\theta(G^{(s)}|G) = \prod_{v \in \mathcal{V}} p_\theta(v \in \mathcal{V}^{(s)}|G)$. In practice, the GNN-based extractor takes $G$ as input and outputs a node probability vector $\mathbf{p} \in \mathbb{R}^{n \times 1}$, where each entry indicates the probability that the corresponding node belongs to $G^{(s)}$. Similarly, for the dual hypergraph view $G^*$, an edge-centric HGNN serves as the subgraph extractor $p_\psi$. It takes $G^*$ as input and outputs an edge probability vector $\mathbf{p}^* \in \mathbb{R}^{m \times 1}$ that indicates if the dual nodes (corresponding to the edges in the original graphs) belong to $G^{*(s)}$. Once the probability vectors are calculated, the subgraph extraction can be executed by:

$$G^{(s)} = (\mathbf{A}, \mathbf{X}^{(s)}) = (\mathbf{A}, \mathbf{X} \odot \mathbf{p}), \quad G^{*(s)} = (\mathbf{M}^*, \mathbf{X}^{*(s)}) = (\mathbf{M}^*, \mathbf{X}^* \odot \mathbf{p}^*), \quad (4)$$

where $\odot$ is the row-wise production. Then, to implement the second term in Eq. (3), we can lift the node probabilities $\mathbf{p}$ to edge probabilities by $\mathbf{p}'$ by $\mathbf{p}'_{\mathbb{I}(e_{ij})} = \mathbf{p}_i \mathbf{p}_j$, where $\mathbb{I}(e_{ij})$ is the index of edge connecting node $v_i$ and $v_j$. After re-probabilizing $\mathbf{p}'$, the SKL divergence between $\mathbf{p}'$ and $\mathbf{p}^*$ can be computed as the regularization term in MSIB framework.

Although the above "two-extractor" design correlates to the theoretical framework of MSIB, in practice, it is non-trivial to ensure the consistency of two generated subgraphs only with an SKL divergence loss. The main reason is that the input and architectures of two extractors are quite different, leading to the difficulty in output alignment. However, the consistency of two bottleneck subgraphs not only guarantees the informativeness of cross-view MI for abnormality measurement, but also affects the quality of explanations. Considering the significance of preserving consistency, we use a single extractor to generate bottleneck subgraphs for two views. In specific, the bottleneck subgraph extractor first takes the original graph $G$ as its input and outputs the node probability vector $\mathbf{p}$ for the bottleneck subgraph extraction of $G^{(s)}$. Then, leveraging the node-edge correspondence in DHT, we can directly lift the node probabilities to edge probability vector $\mathbf{p}^*$ via $\mathbf{p}^*_{\mathbb{I}(e_{ij})} = \mathbf{p}_i \mathbf{p}_j$ and re-probabilization operation. $\mathbf{p}^*$ can be used to extract subgraph for the dual hypergraph view. In this way, the generated bottleneck subgraphs in two views can be highly correlated, enhancing the quality of GLAD prediction (MI) and its explanations. Meanwhile, such a "single-extractor" design further simplifies the model architecture by removing extra extractor and loss function (i.e., the $D_{SKL}$ term in Eq. (3)), reducing the model complexity. Empirical comparison in Sec. 4.4 also validates the effectiveness of this design.

### 3.4 Cross-view MI Maximization

After bottleneck subgraph extraction, the next step is to maximize the MI $I(G^{(s)}; G^{*(s)})$ between the bottleneck subgraphs from two views. The estimated MI, in the testing phase, can be used to evaluate the graph-level abnormality. Owing to the discrete and complex nature of graph-structured data, it is difficult to directly estimate the MI between two subgraphs. Alternatively, a feasible solution is to obtain compact representations for two subgraphs, and then, calculate the representation-level MI as a substitute. In SIGNET, we use message passing-based GNN and HGNN with pooling layer (formulated in Appendix C) to learn the subgraph representations $\mathbf{h}_{G^{(s)}}$ and $\mathbf{h}_{G^{*(s)}}$ for $G^{(s)}$ and $G^{*(s)}$, respectively. In this case, $I(G^{(s)}; G^{*(s)})$ can be transferred into a tractable term, i.e., the MI between subgraph representations $I(\mathbf{h}_{G^{(s)}}; \mathbf{h}_{G^{*(s)}})$.

After that, the MI term $I(\mathbf{h}_{G^{(s)}}; \mathbf{h}_{G^{*(s)}})$ can be maximized by using sample-based differentiable MI lower bounds [37], such as Jensen-Shannon (JS) estimator [54], Donsker-Varadhan (DV) estima-

tor [55], and Info-NCE estimator [56]. Due to its strong robustness and generalization ability [37, 57], we employ Info-NCE for MI estimation in SIGNET. Specifically, given a batch of graph samples $\mathcal{B} = \{G_1, \cdots, G_B\}$, the training loss of SIGNET can be written by:

$$\mathcal{L} = -\frac{1}{2|\mathcal{B}|} \sum_{G_i \in \mathcal{B}} I(\mathbf{h}_{G_i^{(s)}}; \mathbf{h}_{G_i^{*(s)}}) = -\frac{1}{2|\mathcal{B}|} \sum_{G_i \in \mathcal{B}} \left( \ell(\mathbf{h}_{G_i^{(s)}}, \mathbf{h}_{G_i^{*(s)}}) + \ell(\mathbf{h}_{G_i^{*(s)}}, \mathbf{h}_{G_i^{(s)}}) \right),$$

$$\ell(\mathbf{h}_{G_i^{(s)}}, \mathbf{h}_{G_i^{*(s)}}) = \log \frac{exp\left( f_k(\mathbf{h}_{G_i^{(s)}}, \mathbf{h}_{G_i^{*(s)}})/\tau \right)}{\sum_{G_j \in \mathcal{B} \backslash G_i} exp\left( f_k(\mathbf{h}_{G_i^{(s)}}, \mathbf{h}_{G_j^{(s)}})/\tau \right)}, \tag{5}$$

where $f_k(\cdot, \cdot)$ is the cosine similarity function, $\tau$ is the temperature hyper-parameter, and $\ell(\mathbf{h}_{G_i^{*(s)}}, \mathbf{h}_{G_i^{(s)}})$ is calculated following $\ell(\mathbf{h}_{G_i^{(s)}}, \mathbf{h}_{G_i^{*(s)}})$.

### 3.5 Self-Interpretable GLAD Inference

In this subsection, we introduce the self-interpretable GLAD inference protocol with SIGNET (marked in red in Fig. 2(a)) that is composed of two parts: anomaly scoring and explanation.

**Anomaly scoring.** By minimizing Eq. (5) on training data, the cross-view matching patterns of normal samples are well captured, leading to a higher MI for normal data; on the contrary, the anomalies with anomalous attributal and structural characteristics tend to violate the matching patterns, resulting in their lower cross-view MI in our model. Leveraging this property, during inference, the negative of MI can indicate the abnormality of testing data. For a testing sample $G_i$, its anomaly score $s_i$ can be calculated by $s_i = -I(\mathbf{h}_{G_i^{(s)}}; \mathbf{h}_{G_i^{*(s)}})$, where the MI is estimated by Info-NCE.

**Explanation.** In SIGNET, the bottleneck subgraph extractor is able to pinpoint the key substructure of the input graph under the guidance of MSIB framework. The learned bottleneck subgraphs are the most discriminative components of graph samples and are highly related to the anomaly scores. Therefore, we can directly regard the bottleneck subgraphs as the explanations of anomaly detection results. In specific, the node probabilities $\mathbf{p}$ and edge probabilities $\mathbf{p}^*$ can indicate the significance of nodes and edges, respectively. In practical inference, we can pick the nodes/edges with top-k probabilities or use a threshold-based strategy to acquire a fixed-size explanation subgraph $G^{(es)}$.

More discussion about methodology, including the pseudo-code algorithm of SIGNET, the comparison between SIGNET and existing method, and the complexity analysis of SIGNET, is illustrated in Appendix E.

## 4 Experiments

In this section, extensive experiments are conducted to answer three research questions:
- **RQ1:** Can SIGNET provide informative explanations for the detection results?
- **RQ2:** How effective is SIGNET on identifying anomalous graph samples?
- **RQ3:** What are the contributions of the core designs in SIGNET model?

### 4.1 Experimental Setup

**Datasets.** For the explainable GLAD task, we introduce 6 datasets with ground-truth explanations, including three synthetic datasets and three real-world datasets. Details are demonstrated below. We also verify the anomaly detection performance of SIGNET on 10 TU datasets [58], following the setting in [4]. Detailed statistics and visualization of datasets are demonstrated in Appendix F.1.
- **BM-MT, BM-MN, and BM-MS** are three synthetic dataset created by following [13, 19]. Each graph is composed of one base (Tree, Ladder, or Wheel) and one or more motifs that decide the abnormality of the graph. For BM-MT (motif type), each normal graph has a house motif and each anomaly has a 5-cycle motif. For BM-MN (motif number), each normal graph has 1 or 2 house motifs and each anomaly has 3 or 4 house motifs. For BM-MS (motif size), each normal graph has a cycle motif with 3~5 nodes and each anomaly has a cycle motif with 6~9 nodes. The ground-truth explanations are defined as the nodes/edges within motifs.

Table 1: Explanation performance in terms of *NX-AUC* and *EX-AUC* (in percent, mean ± std). The best and runner-up results are highlighted with **bold** and underline, respectively.

| Dataset | Metric | OCGIN-GE | GLocalKD-GE | OCGTL-GE | OCGIN-PG | GLocalKD-PG | OCGTL-PG | SIGNET |
|---|---|---|---|---|---|---|---|---|
| BM-MT | *NX-AUC* | 48.26±3.18 | 49.67±0.88 | 45.79±2.53 | - | - | - | **78.41±6.88** |
|  | *EX-AUC* | 52.03±4.32 | 49.11±2.77 | 49.80±2.88 | 64.08±12.23 | 74.59±7.66 | 72.72±10.19 | **77.69±13.14** |
| BM-MN | *NX-AUC* | 46.25±4.60 | 49.10±0.71 | 40.53±3.18 | - | - | - | **76.57±6.62** |
|  | *EX-AUC* | 60.02±9.20 | 50.17±3.14 | 56.34±3.10 | 54.01±8.01 | 78.68±8.33 | 74.36±12.78 | **83.45±9.33** |
| BM-MS | *NX-AUC* | 52.43±1.70 | 50.43±0.62 | 53.44±1.15 | - | - | - | **76.42±6.81** |
|  | *EX-AUC* | 54.31±9.61 | 49.10±2.29 | 66.87±1.44 | 43.67±12.66 | **82.53±8.56** | 77.45±10.93 | 79.48±9.97 |
| MNIST-0 | *NX-AUC* | 49.48±0.58 | 50.11±0.64 | 38.87±3.21 | - | - | - | **70.38±5.64** |
|  | *EX-AUC* | 50.85±4.77 | 49.75±0.55 | 41.42±2.40 | 39.53±1.51 | 54.69±1.78 | 59.25±4.68 | **72.78±7.25** |
| MNIST-1 | *NX-AUC* | 48.21±2.01 | 49.50±0.50 | 47.04±1.66 | - | - | - | **68.44±3.07** |
|  | *EX-AUC* | 48.60±3.28 | 49.78±0.26 | 45.24±1.11 | 47.98±4.24 | 49.24±1.95 | 57.93±8.54 | **74.83±5.24** |
| MUTAG | *NX-AUC* | 48.99±1.50 | 49.70±1.11 | 49.31±4.94 | - | - | - | **75.60±8.94** |
|  | *EX-AUC* | 51.92±9.05 | 47.65±1.19 | 45.80±2.81 | 46.22±7.90 | 70.47±5.26 | 65.03±16.90 | **78.05±9.19** |

- **MNIST-0 and MNIST-1** are two GLAD datasets derived from MNIST-75sp superpixel dataset [59]. Following [60], we consider a specific class (i.e., digit 0 or 1) as the normal class, and regard the samples belonging to other classes as anomalies. The ground-truth explanations are the nodes/edges with nonzero pixel values.
- **MUTAG** is a molecular property prediction dataset [61]. We set nonmutagenic molecules as normal samples and mutagenic molecules as anomalies. Following [20], -NO2 and -NH2 in mutagenic molecules are viewed as ground-truth explanations.

**Baselines.** Considering their competitive performance, we consider three state-of-the-art deep GLAD methods, i.e., OCGIN [7], GLocalKD [4], and OCGTL [8], as baselines. To provide explanations for them, we integrate two mainstream post-hoc GNN explainers, i.e., GNNExplainer [19] (GE for short) and PGExplainer [20] (PG for short) into the deep GLAD methods. For GLAD tasks, we further introduce the baselines composed of a graph kernel (i.e., Weisfeiler-Lehman kernel (WL) [62] or Propagation kernel (PK) [63]) and a detector (i.e., iForest (iF) [64] or one-class SVM (OCSVM) [65]).

**Metrics and Implementation.** For interpretation evaluation, we report explanation ROC-AUCs at node level (NX-AUC) and edge level (EX-AUC) respectively, similar to [19, 20]. For GLAD performance, we report the ROC-AUC w.r.t. anomaly scores and labels (AD-AUC) [7]. We repeat 5 times for all experiments and record the average performance. In SIGNET, we use GIN [2] and Hyper-Conv [30] as the GNN and HGNN encoders. The bottleneck subgraph extractor is selected from GIN [2] and MLP. We perform grid search to pick the key hyper-parameters in SIGNET and baselines. More details of implementation and infrastructures are in Appendix F. Our code is available at https://github.com/yixinliu233/SIGNET.

## 4.2 Explainability Results (RQ1)

**Quantitative evaluation.** In Table 1, we report the node-level and edge-level explanation AUC [22] on 6 datasets. Note that PGExplainer [20] can only provide edge-level explanations natively. We have the following observations: 1) *SIGNET achieves SOTA performance in almost all scenarios.* Compared to the best baselines, the average performance gains of SIGNET are 27.89% in NX-AUC and 8.99% in EX-AUC. The superior performance verifies the significance of learning to interpret and detect with a unified model. 2) *The post-hoc explainers are not compatible with all GLAD models.* For instance, PGExplainer works relevantly well with GLocalKD but cannot provide informative explanations for OCGIN. The GN-NExplainer, unfortunately, exhibits poor performance in most scenarios. 3) *SIGNET has larger performance gains on real-world datasets*, which illustrates the potential of SIGNET in explaining real-world GLAD tasks. 4) Despite its superior performance, *the stability of SIGNET is relevantly average*. Especially on the synthetic datasets, we can find that the standard deviations of

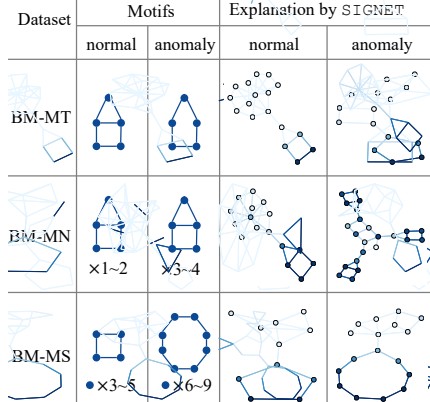

Figure 3: Visualization of explanation results w.r.t. node and edge probabilities.

Table 2: Anomaly detection performance in terms of *AD-AUC* (in percent, mean ± std). The best and runner-up results are highlighted with **bold** and underline, respectively.

| Dataset | PK-OCSVM | PK-iF | WL-OCSVM | WL-iF | OCGIN | GLocalKD | OCGTL | SIGNET |
|---|---|---|---|---|---|---|---|---|
| BM-MT | $52.58_{\pm0.35}$ | $45.30_{\pm1.60}$ | $53.36_{\pm0.46}$ | $50.30_{\pm0.40}$ | $73.33_{\pm6.18}$ | $74.94_{\pm5.12}$ | $93.61_{\pm0.20}$ | $\mathbf{95.89_{\pm2.75}}$ |
| BM-MN | $97.13_{\pm0.19}$ | $56.80_{\pm4.65}$ | $76.60_{\pm0.77}$ | $49.90_{\pm0.20}$ | $59.35_{\pm2.81}$ | $77.51_{\pm3.08}$ | $\mathbf{99.49_{\pm0.05}}$ | $93.41_{\pm1.66}$ |
| BM-MS | $79.34_{\pm0.52}$ | $57.00_{\pm6.15}$ | $56.19_{\pm0.42}$ | $51.30_{\pm0.03}$ | $58.00_{\pm3.44}$ | $65.03_{\pm2.36}$ | $92.01_{\pm0.82}$ | $\mathbf{94.01_{\pm4.88}}$ |
| MNIST-0 | $48.89_{\pm2.19}$ | $59.58_{\pm1.50}$ | $67.19_{\pm2.69}$ | $59.27_{\pm2.43}$ | $69.54_{\pm2.61}$ | $82.29_{\pm1.65}$ | $80.68_{\pm3.14}$ | $\mathbf{83.25_{\pm2.17}}$ |
| MNIST-1 | $43.45_{\pm1.32}$ | $78.97_{\pm5.54}$ | $63.64_{\pm2.60}$ | $65.05_{\pm3.21}$ | $\mathbf{98.25_{\pm0.61}}$ | $93.04_{\pm0.65}$ | $97.98_{\pm0.36}$ | $90.12_{\pm5.21}$ |
| MUTAG | $53.30_{\pm1.29}$ | $43.60_{\pm1.72}$ | $49.64_{\pm2.89}$ | $49.07_{\pm0.32}$ | $57.59_{\pm3.36}$ | $65.09_{\pm5.94}$ | $63.41_{\pm2.60}$ | $\mathbf{87.72_{\pm3.48}}$ |
| PROTEINS-F | $50.49_{\pm4.92}$ | $60.70_{\pm2.55}$ | $51.35_{\pm4.35}$ | $61.36_{\pm2.54}$ | $70.89_{\pm2.44}$ | $\mathbf{77.30_{\pm5.15}}$ | $76.51_{\pm1.55}$ | $75.22_{\pm3.91}$ |
| ENZYMES | $53.67_{\pm2.66}$ | $51.30_{\pm2.01}$ | $55.24_{\pm2.66}$ | $51.60_{\pm3.81}$ | $58.75_{\pm5.98}$ | $61.39_{\pm8.81}$ | $62.06_{\pm3.36}$ | $\mathbf{62.96_{\pm4.22}}$ |
| AIDS | $50.79_{\pm4.30}$ | $51.84_{\pm2.87}$ | $50.12_{\pm3.43}$ | $61.13_{\pm0.71}$ | $78.16_{\pm3.05}$ | $93.27_{\pm4.19}$ | $\mathbf{99.40_{\pm0.57}}$ | $97.27_{\pm1.17}$ |
| DHFR | $47.91_{\pm3.76}$ | $52.11_{\pm3.96}$ | $50.24_{\pm3.13}$ | $50.29_{\pm2.77}$ | $49.23_{\pm3.05}$ | $56.71_{\pm3.57}$ | $59.90_{\pm2.96}$ | $\mathbf{74.01_{\pm4.69}}$ |
| BZR | $46.85_{\pm5.31}$ | $55.32_{\pm6.18}$ | $50.56_{\pm5.87}$ | $52.46_{\pm3.30}$ | $65.91_{\pm1.47}$ | $69.42_{\pm7.78}$ | $63.94_{\pm8.89}$ | $\mathbf{81.44_{\pm9.23}}$ |
| COX2 | $50.27_{\pm7.91}$ | $50.05_{\pm2.06}$ | $49.86_{\pm7.43}$ | $50.27_{\pm0.34}$ | $53.58_{\pm5.05}$ | $59.37_{\pm12.67}$ | $55.23_{\pm5.68}$ | $\mathbf{71.46_{\pm4.64}}$ |
| DD | $48.30_{\pm3.98}$ | $71.32_{\pm2.41}$ | $47.99_{\pm4.09}$ | $70.31_{\pm1.09}$ | $72.27_{\pm1.83}$ | $\mathbf{80.12_{\pm5.24}}$ | $79.48_{\pm2.02}$ | $72.72_{\pm3.91}$ |
| NCI1 | $49.90_{\pm1.18}$ | $50.58_{\pm1.38}$ | $50.63_{\pm1.22}$ | $50.74_{\pm1.70}$ | $71.98_{\pm1.21}$ | $68.48_{\pm2.39}$ | $73.44_{\pm0.97}$ | $\mathbf{74.89_{\pm2.07}}$ |
| IMDB-B | $50.75_{\pm3.10}$ | $50.80_{\pm3.17}$ | $54.08_{\pm5.19}$ | $50.20_{\pm0.40}$ | $60.19_{\pm8.90}$ | $52.09_{\pm3.41}$ | $64.05_{\pm3.32}$ | $\mathbf{66.48_{\pm3.49}}$ |
| REDDIT-B | $45.68_{\pm2.24}$ | $46.72_{\pm3.42}$ | $49.31_{\pm2.33}$ | $48.26_{\pm0.32}$ | $75.93_{\pm8.65}$ | $77.85_{\pm2.62}$ | $\mathbf{86.81_{\pm2.10}}$ | $82.78_{\pm1.11}$ |
| Avg. Rank | 6.6 | 6.1 | 6.2 | 6.4 | 4.1 | 2.8 | 2.1 | **1.7** |

NX-AUC and EX-AUC are large. We speculate that the instability is due to the lack of labels that provide reliable supervisory signals for anomaly detection explanations.

**Qualitative evaluation.** To better understand the behavior of SIGNET, we visualize the explanations (i.e., node and edge probabilities) in Fig. 3. We can witness that SIGNET can assign larger probabilities for the nodes and edges within the discriminative motifs, providing valid explanations for the GLAD predictions. In contrast, the probabilities of base subgraphs are uniformly small, indicating that SIGNET is able to ignore the unrelated substructure. However, we can still witness some irrelevant nodes included in the explanations in the anomaly sample of BM-MN dataset, which indicates that SIGNET may generate noisy explanations in some special cases.

## 4.3 Anomaly Detection Results (RQ2)

To investigate the anomaly detection performance of SIGNET, we conduct experiments on 16 datasets and summarize the results in Table 2. The following observations can be concluded: 1) *SIGNET outperforms all baselines on 10 datasets and achieves competitive performance on the rest.* The main reason is that SIGNET can capture graph patterns at node and edge level with two distinct views and concentrate on the key substructure during anomaly scoring. 2) *The deep GLAD methods generally perform better than kernel-based methods*, indicating that GNN-based deep learning models are effective in identifying anomalous graphs. To sum up, SIGNET can not only accurately detect anomalies but also provide informative interpretations for the predictions.

## 4.4 Ablation Study (RQ3)

We perform ablation studies to evaluate the effectiveness of core designs in SIGNET, i.e., dual hypergraph-based view construction, single extractor, and InfoNCE MI estimator. We replace these components with alternative designs, and the results are illustrated in Table 3. We consider two strategies for *view construction*: augmentation-based view construction [46] (Aug. View) and structural property-based view construction [50] (Str. View). As we discussed in Sec. 3.2, perturbation-based

Table 3: Performance of SIGNET and its variants.

| Variant | BM-MS | | | MNIST-0 | | |
|---|---|---|---|---|---|---|
| | *NX-AUC* | *EX-AUC* | *AD-AUC* | *NX-AUC* | *EX-AUC* | *AD-AUC* |
| Aug. View | 66.87 | 64.48 | 63.26 | 41.82 | 39.25 | 48.75 |
| Str. View | 70.36 | 68.27 | 92.79 | 51.73 | 50.40 | 55.29 |
| 2E w $D_{SKL}$ | 70.71 | 65.23 | 92.44 | 59.22 | 57.33 | 81.77 |
| 2E w/o $D_{SKL}$ | 73.54 | 37.15 | 87.84 | 57.92 | 43.65 | 80.97 |
| JS MI Est. | 70.35 | 68.96 | 75.21 | 59.06 | 63.85 | 71.78 |
| DV MI Est. | 72.53 | 74.16 | 74.87 | 60.90 | 70.28 | 70.31 |
| SIGNET | 76.42 | 79.48 | 94.01 | 70.38 | 72.78 | 83.25 |

graph augmentation is not appropriate for GLAD tasks, leading to its poor performance. The structural property-based strategy has better performance on BM-BT but still performs weakly on MNIST-0. In contrast, dual hypergraph-based view construction is a better strategy that jointly considers node-level and edge-level information, contributing to optimal detection and explanation performance. We also test the performance of SIGNET with two extractors (2E) and discuss the contribution of SKL divergence. We can find that even with the help of $D_{SKL}$, the two-extractor version still underperforms

the original `SIGNET` with one extractor. Meanwhile, we can witness that compared to JS [54] and DV [55] MI estimators, Info-NCE estimator can lead to superior performance, especially for anomaly detection.

## 5 Conclusion

This paper presents a novel and practical research problem, explainable graph-level anomaly detection (GLAD). Based on the information bottleneck principle, we deduce the framework multi-view subgraph information bottleneck (MSIB) to address the explainable GLAD problem. We develop a new method termed `SIGNET` by instantiating MSIB framework with advanced neural modules. Extensive experiments verify the effectiveness of `SIGNET` in identifying anomalies and providing explanations. A limitation of our paper is that we mainly focus on purely unsupervised GLAD scenarios where ground-truth labels are entirely unavailable. As a result, for few-shot or semi-supervised GLAD scenarios [44] where a few labels are accessible, `SIGNET` cannot directly leverage them for model training and self-interpretation. We leave the exploration of supervised/semi-supervised self-interpretable GLAD problems in future works.

## Acknowledgment

S. Pan was partially supported by an Australian Research Council (ARC) Future Fellowship (FT210100097). F. Li was supported by the National Natural Scientific Foundation of China (No. 62202388) and the National Key Research and Development Program of China (No. 2022YFF1000100).

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
