# Supplementary Material for "Towards Self-Interpretable Graph-Level Anomaly Detection"

**Yixin Liu[1], Kaize Ding[2], Qinghua Lu[3], Fuyi Li[4,5], Leo Yu Zhang[6], Shirui Pan[6]***
[1]Monash University, [2]Northwestern University, [3]Data61, CSIRO,
[4]Northwest A&F University, [5]The University of Adelaide, [6]Griffith University
yixin.liu@monash.edu, kaize.ding@northwestern.edu, qinghua.lu@data61.csiro.au,
fuyi.li@nwsuaf.edu.cn, leo.zhang@griffith.edu.au, s.pan@griffth.edu.au

## A  Definitions of "Explainability" and "Interpretability"

Since explainable artificial intelligence is an emerging area of research, how to specifically discriminate similar concepts "explainability" and "interpretability" is not yet completely standardized. Following the recent survey paper [1], we distinguish them with definite principles rather than using them interchangeably.

Specifically, we define the term "explainability" as a more general and high-level concept that includes all learning scenarios, models, and strategies related to providing understandable knowledge for the predictions. The major reason is that "explainable artificial intelligence" and "explainable machine learning" are well-known concepts in the community. For instance, we denote the ability to explain GNNs' predictions as "explainability of GNNs", and the related learning tasks include explainable node classification, explainable graph classification, etc. Following this way, we denote our proposed learning problem as "explainable graph-level anomaly detection (GLAD)".

Differently, we denote "interpretability" as the ability of a model to intrinsically provide explanations for itself. To well emphasis the characteristic of interpreting itself, we sometimes use the concept "self-interpretability" interchangeably with the concept "interpretability". For instance, the GNNs that can jointly generate predictions and explanations are denoted as "interpretable GNNs" or "self-interpretable GNNs". Under such a definition, the models that provide post-hoc explanations for trained GNNs are not interpretable. In this paper, we aim to propose a "self-interpretable GLAD model" that is able to yield explanations for the anomaly detection results by itself.

## B  Related Work in Detail

**Graph Neural Networks (GNNs).** GNNs are the extension of the convolution-based neural networks onto graph data [2, 3, 4, 5, 6, 7, 8, 9]. Early GNNs define graph convolution based on spectral theory [8, 10]. Recently, the mainstream GNNs usually follow the paradigm of message passing for spatial graph convolution, i.e., executing graph convolution by aggregating the information for adjacent nodes [3, 4, 5, 6]. For instance, GCN [4] uses an average-based aggregation function for message passing. GIN [3], differently, employs a summation-based aggregation function to ensure its expressive ability. Apart from normal GNNs designed for simple graphs where each edge connects exactly two nodes, some recent studies apply GNNs to hypergraphs, a generalization of graphs where an edge can connect more than two nodes [11, 12, 13]. Among them, Hyper-Conv [12] is a representative HGNN that applies a GCN-like aggregation function to the graph convolution for hypergraphs. Thanks to their strong expressive power, GNNs are effective in various graph learning tasks, such as node classification [4, 14, 15, 16], graph classification [3], and also graph anomaly detection [17, 18]. Besides, GNNs can also be widely applied to diverse real-world learning

---

*Corresponding Author.

scenarios, such as federated learning [19, 20], knowledge graph reasoning [21, 22, 23], adversarial attack [24, 25], and molecule analysis [26, 27].

**Explainability of GNNs.** To make the predictions of GNNs transparent and understandable, a line of studies proposes to uncover the explanation, i.e., the critical subgraphs and/or features that highly correlate to the prediction, for GNN models [1, 28, 29, 30, 31, 32]. Existing methods can be divided into two types: post-hoc GNN explainer and self-interpretable GNN [30, 32]. The post-hoc GNN explainers use specialized models or strategies to explain the behavior of a trained GNN, such as input perturbation [28, 29], surrogate model [31], and prediction decomposition [33]. For instance, PGExplainer [29] uses an edge embedding-based neural module to modify the input graph, and the learning objective is to optimize the cross-entropy between the original prediction and the modified input. Differently, the self-interpretable GNNs can intrinsically provide explanations for the predictions using the interpretable designs in GNN architectures [34, 30, 35]. GSAT [35] is one of the self-interpretable GNNs that uses a parameterized attention module to pick the graph rationale along with the training of the GNN backbone. Theoretically, the post-hoc explainers can be used to explain the well-trained GLAD models; however, the post-hoc explainers can potentially provide sub-optimal solutions since they are not directly learned with the detection models. On the other hand, most self-interpretable GNNs are designed to explain the prediction of supervised tasks, especially graph/node classification tasks. In this case, it is non-trivial to directly apply them to unsupervised graph anomaly detection tasks, since their inherent supervised learning objective cannot work without ground-truth labels.

**Graph Anomaly Detection.** The objective of graph anomaly detection is to identify anomalies that deviate from the majority of samples in graph-structured data [36, 17, 37]. Most efforts mainly focus on node-level anomaly detection, i.e., detecting the abnormal nodes from one or more graphs [17, 18, 38, 39]. In this paper, we mainly investigate graph-level anomaly detection (GLAD) that aims to recognize anomalous graphs from a set of graphs [36, 40, 41, 42]. A few recent studies try to address the GLAD problem with various advanced techniques. For example, OCGIN [36] combines the objective of one-class classification and a GIN encoder into the first GLAD model. GLocalKD [40] uses the knowledge distillation error between a random network and a trainable network to evaluate the abnormality of graph samples. OCGTL [41] introduces a graph transformation learning-based learning objective to identify the anomalous samples in a graph set. However, these methods can only predict the scores to indicate the degree of abnormality of each sample, but cannot provide the behind explanations, i.e., the substructure causes the abnormality. To boost the reliability and explainability of GAD methods, in this paper, we propose a self-interpretable GAD framework to generate both anomaly prediction and its explanation.

**Learning by Information-Bottleneck (IB).** IB is an information theory-based approach for representation learning that trains the encoder by preserving the information that is relevant to label prediction while minimizing the amount of superfluous information [43, 44, 45]. Formally, the objective of IB principle is to maximize the mutual information (MI) between representation $Z$ and label $Y$, and minimize the MI between representation $Z$ and original data $X$ [45]. Some pioneering efforts [46, 47] extend IB principle to multi-view learning scenarios, and some of them enable the application of IB principle for unsupervised learning [46]. Recent efforts also attempt to apply IB principle to graph learning tasks [48, 35, 49, 50, 51, 52, 53, 54]. One feasible idea is to borrow the representation-based IB principle for graph representation learning [50, 53]; another line of work regards a vital bottleneck subgraph $G^{(s)}$ rather than the representation $Z$ as the bottleneck and tries to maximize the MI between $G^{(s)}$ and label $Y$ while minimizing the MI between $G^{(s)}$ and original graph $G$ [48, 51, 52].

**Explainable anomaly detection.** Anomaly detection is an essential machine learning task that aims to detect unusual or rare patterns or instances within a dataset [55, 56]. In order to improve the trustworthiness and comprehensibility of anomaly detection systems, a brunch of research termed explainable anomaly detection focuses on generating valid explanations for the results given by anomaly detection models [57, 58, 59]. For example, to provide explanations for one-class image anomaly detection models, FCDD [57] uses a fully convolutional module to generate pixel-level explanation. ATON [58] utilizes an attention-guided triplet deviation mechanism to provide explanations for any black-box outlier detector on tabular data. Cho et al. [59] introduce an auxiliary prototypical classifier to learn explanations of anomaly detection models for medical images. Despite their success, these techniques cannot be directly applied to graph-structured data.

## C  Formulations of GNN and HGNN

In this section, we provide detailed definitions of message passing-based graph neural network (GNN) and hypergraph neural network (HGNN). Given a simple graph $G$, the target of a GNN is to learn the node-level representation following the message passing scheme:

$$\mathbf{h}_v^{(l+1)} = \text{UPDATE}\left(\mathbf{h}_v^{(l)}, \text{AGGREGATE}\left(\left\{\mathbf{h}_u^{(l)} : \forall u \in \mathcal{N}(v; \mathbf{A})\right\}\right)\right), \qquad \text{(A.1)}$$

where $\mathbf{h}_v^{(l)}$ is the latent representation vector for node $v \in \mathcal{V}$ at the $l$-th layer (with $\mathbf{h}_v^{(0)} = \mathbf{x}_v = \mathbf{X}_{[v]}$), $\mathcal{N}(v; \mathbf{A})$ is the neighboring node set of $v$ obtained from $\mathbf{A}$, AGGREGATE$(\cdot)$ is is the function that aggregates messages from neighboring nodes, and UPDATE$(\cdot)$ is the function that updates the node representation. With similar notations, we can formulate a HGNN as:

$$\mathbf{h}_{v^*}^{(l+1)} = \text{UPDATE}\left(\mathbf{h}_{v^*}^{(l)}, \text{AGGREGATE}\left(\left\{\mathbf{h}_{u^*}^{(l)} : \forall u^* \in \mathcal{N}(v^*; \mathbf{M}^*)\right\}\right)\right), \qquad \text{(A.2)}$$

where $\mathcal{N}(v^*; \mathbf{M}^*)$ is the neighboring node set of $v^* \in \mathcal{V}^*$ obtained from incidence matrix $\mathbf{M}^*$.

In GNNs, a pooling operation POOL$(\cdot)$ can be applied to obtain a graph-level representation vector with $\mathbf{h}_G = \text{POOL}\left(\{\mathbf{h}_v^{(L)} : \forall v \in \mathcal{V}\}\right)$ by summarizing the representations of all nodes at the final layer $L$. A similar pooling layer can be used to obtain hypergraph-level representation $\mathbf{h}_G^*$.

## D  MSIB Loss Computation

Starting from Eq. (2), we can first rewrite the objective of the first graph view $G^1$ as a loss function into:

$$\mathcal{L}_1 = I(G^1; G^{1(s)}|G^2) - \frac{1}{\beta_1}I(G^2; G^{1(s)}), \qquad \text{(A.3)}$$

which we aim to minimize during model training. Similar to Eq. (A.3), the corresponding loss function for the second graph view $G^2$ can be written by:

$$\mathcal{L}_2 = I(G^2; G^{2(s)}|G^1) - \frac{1}{\beta_2}I(G^1; G^{2(s)}), \qquad \text{(A.4)}$$

where $\beta_2$ is the trade-off parameter for $\mathcal{L}_2$. Then, by computing the average of $\mathcal{L}_1$ and $\mathcal{L}_2$, we have a joint loss function to optimize both $G^{1(s)}$ and $G^{2(s)}$:

$$\mathcal{L}_{joint} = \frac{I(G^1; G^{1(s)}|G^2) + I(G^2; G^{2(s)}|G^1)}{2} - \frac{\frac{1}{\beta_1}I(G^2; G^{1(s)}) + \frac{1}{\beta_2}I(G^1; G^{2(s)})}{2}. \qquad \text{(A.5)}$$

For term $I(G^1; G^{1(s)}|G^2)$, we can derive its upper bound by:

$$
\begin{aligned}
I_\theta\left(G^1; G^{1(s)}|G^2\right) &= \mathbb{E}_{\boldsymbol{G}^1, \boldsymbol{G}^2 \sim p\left(G^1, G^2\right)} \mathbb{E}_{\boldsymbol{G}^{(s)} \sim p_\theta\left(G^{1(s)}|G^1\right)} \left[\log \frac{p_\theta\left(G^{1(s)} = \boldsymbol{G}^{(s)}|G^1 = \boldsymbol{G}^1\right)}{p_\theta\left(G^{1(s)} = \boldsymbol{G}^{(s)}|G^2 = \boldsymbol{G}^2\right)}\right] \\
&= \mathbb{E}_{\boldsymbol{G}^1, \boldsymbol{G}^2 \sim p\left(G^1, G^2\right)} \mathbb{E}_{\boldsymbol{G}^{(s)} \sim p_\theta\left(G^{1(s)}|G^1\right)} \left[\log \frac{p_\theta\left(G^{1(s)} = \boldsymbol{G}^{(s)}|G^1 = \boldsymbol{G}^1\right)}{p_\psi\left(G^{2(s)} = \boldsymbol{G}^{(s)}|G^2 = \boldsymbol{G}^2\right)} \frac{p_\psi\left(G^{2(s)} = \boldsymbol{G}^{(s)}|G^2 = \boldsymbol{G}^2\right)}{p_\theta\left(G^{1(s)} = \boldsymbol{G}^{(s)}|G^2 = \boldsymbol{G}^2\right)}\right] \\
&= D_{KL}\left(p_\theta(G^{1(s)}|G^1)\|p_\psi(G^{2(s)}|G^2)\right) - D_{KL}\left(p_\theta(G^{2(s)}|G^1)\|p_\psi(G^{2(s)}|G^2)\right) \\
&\leq D_{KL}\left(p_\theta(G^{1(s)}|G^1)\|p_\psi(G^{2(s)}|G^2)\right).
\end{aligned}
$$

$$\text{(A.6)}$$

where $D_{KL}(\cdot)$ is the Kullback–Leibler (KL) divergence. Analogously, we can acquire the upper bound of $I(G^2; G^{2(s)}|G^1)$ as $D_{KL}\left(p_\theta(G^{2(s)}|G^2)\|p_\psi(G^{1(s)}|G^1)\right)$. In this way, the first term in Eq. (A.5) can be upperbound by:

$$\frac{I(G^1; G^{1(s)}|G^2) + I(G^2; G^{2(s)}|G^1)}{2} \leq D_{SKL}\left(p_\theta(G^{1(s)}|G^1)\|p_\psi(G^{2(s)}|G^2)\right), \quad (A.7)$$

where $D_{SKL}\left(p_\theta(G^{1(s)}|G^1)\|p_\psi(G^{2(s)}|G^2)\right) = \frac{1}{2}D_{KL}\left(p_\theta(G^{1(s)}|G^1)\|p_\psi(G^{2(s)}|G^2)\right) + \frac{1}{2}D_{KL}\left(p_\theta(G^{2(s)}|G^2)\|p_\psi(G^{1(s)}|G^1)\right)$.

Then, according to the chain rule of mutual information, i.e., $I(xy; z) = I(y; z) + I(x; z|y)$, we can reform the term $I(G^2; G^{1(s)})$ by:

$$\begin{aligned}
I(G^{1(s)}; G^2) &= I(G^{1(s)}; G^{2(s)}G^2) - I(G^{1(s)}; G^{2(s)}|G^2) \\
&\overset{(H)}{=} I(G^{1(s)}; G^{2(s)}G^2) \\
&= I(G^{1(s)}; G^{2(s)}) + I(G^{1(s)}; G^2|G^{2(s)}) \\
&\geq I(G^{1(s)}; G^{2(s)}),
\end{aligned} \quad (A.8)$$

where $(H)$ indicates the hypothesis that $G^{2(s)}$ is sufficient for $G^1$, i.e., $I(G^{1(s)}; G^{2(s)}|G^1) = 0$. Symmetrically, we can also have $I(G^{2(s)}; G^1) \geq I(G^{1(s)}; G^{2(s)})$. In this case, the second term in Eq. (A.5) has the lower bound with:

$$\frac{\frac{1}{\beta_1}I(G^2; G^{1(s)}) + \frac{1}{\beta_2}I(G^1; G^{2(s)})}{2} \geq \frac{(\beta_1 + \beta_2)}{2\beta_1\beta_2}I(G^{1(s)}; G^{2(s)}). \quad (A.9)$$

By jointly considering Eq. (A.7) and Eq. (A.9), the joint loss function (Eq. (A.5)) can be bounded by:

$$\mathcal{L}_{joint} \leq D_{SKL}\left(p_\theta(G^{1(s)}|G^1)\|p_\psi(G^{2(s)}|G^2)\right) - \frac{(\beta_1 + \beta_2)}{2\beta_1\beta_2}I(G^{1(s)}; G^{2(s)}). \quad (A.10)$$

Finally, by multiplying both terms with $\beta = \frac{2\beta_1\beta_2}{(\beta_1+\beta_2)}$ and re-parametrizing the objective, we have a tractable loss function for MSIB framework:

$$\mathcal{L}_{MSIB} = -I(G^{1(s)}; G^{2(s)}) + \beta D_{SKL}\left(p_\theta(G^{1(s)}|G^1)\|p_\psi(G^{2(s)}|G^2)\right). \quad (A.11)$$

# E   Methodology Discussion

## E.1   Algorithm

The overall algorithm of `SIGNET` is summarized in Algo. 1.

## E.2   Discussion of `SIGNET` v.s. GSAT

In this paragraph, we discuss the connections and differences between `SIGNET` and the representative self-interpretable GNNs, GSAT.

**Connections between `SIGNET` and GSAT:**

- Theoretical foundation. Both GSAT and `SIGNET` are based on the well-known information theory criteria, the information bottleneck, serving as their theoretical foundation for their explanation target.
- Explanation goal. As an explainable method for graphs, they have a common objective of identifying the key subgraph within the input graph sample that holds the highest relevance to the final prediction.

**Algorithm 1:** The overall algorithm of `SIGNET`

---

**Input:** Training Set $\mathcal{G}_{tr}$; Test Set $\mathcal{G}_{te}$.
**Parameters :** Number of epoch $E$.
**Output:** Anomaly Score Set $\mathcal{S}$; Explanation Subgraph Set $\mathcal{G}^{(es)}$.

   /* Training                                                                                  */

1   Initialize model parameters
2   **for** $e = 1, 2, \cdots, E$ **do**
3      $\mathcal{B}_1, \cdots, \mathcal{B}_{n_b} \leftarrow$ Randomly split $\mathcal{G}_{tr}$ into batches
4      **for** $\mathcal{B} = \mathcal{B}_1, \cdots, \mathcal{B}_{n_b}$ **do**
5         **for** $G_i \in \mathcal{B}$ **do**
6            $G_i^* \leftarrow$ Obtain the dual hypergraph of $G_i$ by DHT
7            $\mathbf{p}_i, \mathbf{p}_i^* \leftarrow$ Calculate probability vectors by neural extractor
8            $G_i^{(s)}, G_i^{*(s)} \leftarrow$ Extract bottleneck subgraphs by Eq. (4)
9            $\mathbf{h}_i^{(s)}, \mathbf{h}_i^{*(s)} \leftarrow$ Calculate graph-level representations by GNN/HGNN encoders
10         **end**
11         $\mathcal{L} \leftarrow$ Calculate Info-NCE loss by Eq. (5)
12         Update model parameters via gradient descent w.r.t. $\mathcal{L}$
13      **end**
14 **end**

   /* Inference                                                                         */

15 **for** $G_i \in \mathcal{G}_{te}$ **do**
16      $G_i^* \leftarrow$ Obtain the dual hypergraph of $G_i$ by DHT
17      $\mathbf{p}_i, \mathbf{p}_i^* \leftarrow$ Calculate probability vectors by neural extractor
18      $G_i^{(s)}, G_i^{*(s)} \leftarrow$ Extract bottleneck subgraphs by Eq. (4)
19      $\mathbf{h}_i^{(s)}, \mathbf{h}_i^{*(s)} \leftarrow$ Calculate graph-level representations by GNN/HGNN encoders
20      $s_i = -I(\mathbf{h}_i^{(s)}, \mathbf{h}_i^{*(s)}) \leftarrow$ Calculate the anomaly score of $G_i$ by Info-NCE MI estimator
21      $G_i^{(es)} \leftarrow$ Extract explanation subgraph according to $\mathbf{p}_i$ and $\mathbf{p}_i^*$ using top-k/threshold strategy
22 **end**
23 $\mathcal{S}, \mathcal{G}^{(es)} \leftarrow$ Collect all the anomaly scores $s_i$ and explanations $G_i^{(es)}$ into sets

---

- Architecture. Both GSAT and `SIGNET` adopt learnable neural networks to parameterize the graph data and make the explanation differentiable, which is a common design among explainable GNNs. However, GSAT only conducts the relaxation at the edge level, while `SIGNET` can provide explanation scores at both node and edge levels.

**Differences between `SIGNET` and GSAT:**

- Targeted tasks. GSAT focuses on a supervised graph-level classification task where categorical labels are available for training the self-interpretation module. On the other hand, `SIGNET` targets unsupervised graph-level anomaly detection, a more challenging task with unavailable labels during training.

- Theoretical framework. GSAT is designed based on the original information bottleneck framework with subgraph bottleneck, tailored to its targeted supervised setting. In contrast, `SIGNET` is based on the multi-view subgraph information bottleneck (MSIB) framework derived in this paper, specifically designed for unsupervised anomaly detection tasks.

- Learning objectives. GSAT is trained using cross-entropy loss, a commonly used classification loss. In contrast, `SIGNET` is optimized using an Info-NCE loss, aiming to maximize the mutual information between each graph and its rational subgraph.

- Graph view for learning. GSAT only considers the original view for graph learning, while `SIGNET` takes both the original and DHT views into account for self-interpretable graph learning.

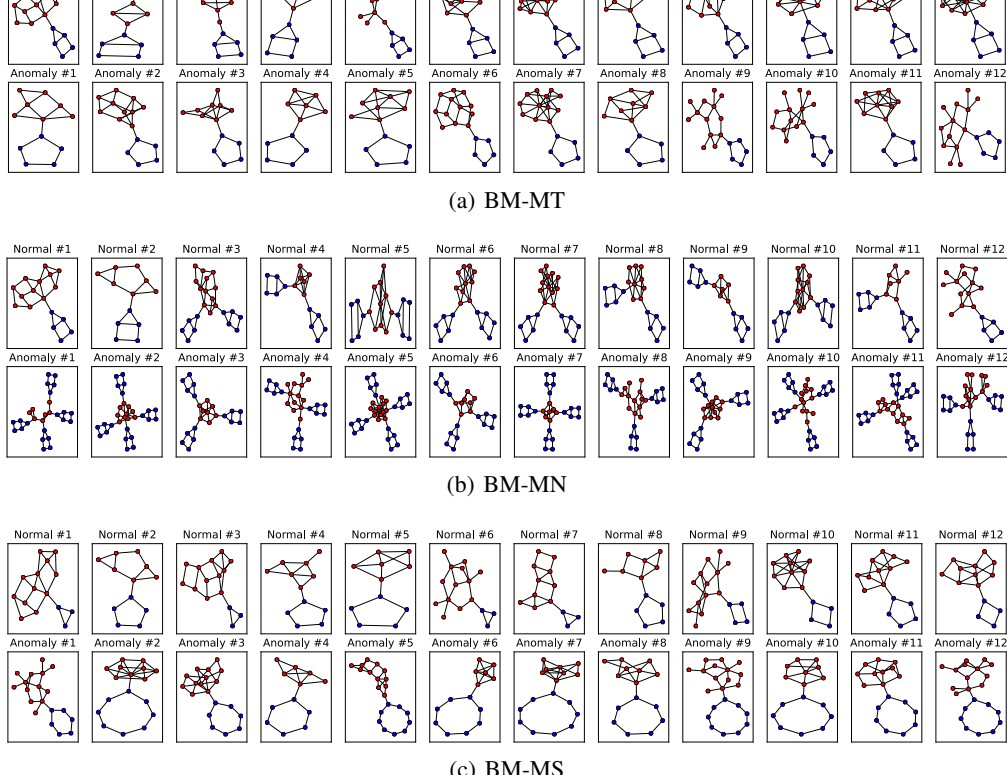

(a) BM-MT

(b) BM-MN

(c) BM-MS

Figure 1: Examples of three synthetic datasets, where subgraphs in blue are the ground-truth explanations.

## E.3 Complexity analysis

Within this paragraph, we denote the average numbers of nodes and edges as $n$ and $m$ respectively, and denote the number of graphs and batch size as $N$ and $B$ respectively. At each training iteration, we first conduct DHT to obtain the dual hypergraph, which requires $\mathcal{O}(N(m+n))$. Then, the GNN-based extractor that calculates probability consumes $\mathcal{O}(NL_1md_1+NL_1nd_1^2+Nnd_1d_f)$ complexity, where $L_1$ and $d_1$ are the layer number and latent dimension of the extractor, respectively. The bottleneck subgraph extraction for two views requires $\mathcal{O}(N(m+n))$ in total. For the GNN and HGNN encoders, their time complexities are $\mathcal{O}(NL_2md_2 + NL_2nd_2^2 + Nnd_2d_f)$ and $\mathcal{O}(NL_2nd_2 + NL_2md_2^2 + Nnd_2d_{f*})$ respectively, where $L_2$ and $d_2$ denote their layer number and latent dimension. Finally, the Info-NCE loss requires $\mathcal{O}(NBd_2)$ complexity. To simplify the overall complexity, we denote the larger terms within $L_1$ and $L_2$ as $L$, and the larger terms between $d_1$ and $d_2$ as $d$. After ignoring the smaller terms, the overall complexity of SIGNET is $\mathcal{O}(NLd^2(m+n) + Nnd(d_f + d_{f*}) + NBd)$.

## F    Supplement of Experimental Setup

### F.1    Datasets

We consider 16 benchmark datasets in total. The statistic of the datasets is provided in Table 1. In this paper, we take "PROTEINS-F", "IMDB-B", and "REDDIT-B" as the abbreviations of "PROTEIN-full", "IMDB-BINARY", and "REDDIT-BINARY", respectively. For our synthetic datasets, we provide some examples in Fig. 1.

Table 1: Statistics of datasets.

| Dataset | # Graphs (Train/Test) | # Nodes (avg.) | # Edges (avg.) |
|---|---|---|---|
| BM-MT | 500/200 | 14.3 | 44.5 |
| BM-MN | 500/200 | 18.4 | 56.7 |
| BM-MS | 500/200 | 14.0 | 42.8 |
| MNIST-0 | 1000/500 | 69.4 | 572.2 |
| MNIST-1 | 1000/500 | 57.9 | 419.6 |
| MUTAG | 1742/295 | 30.1 | 60.9 |
| PROTEINS-F | 360/223 | 39.1 | 72.8 |
| ENZYMES | 400/120 | 32.6 | 62.1 |
| AIDS | 1280/400 | 15.7 | 16.2 |
| DHFR | 368/152 | 42.4 | 44.5 |
| BZR | 69/81 | 35.8 | 38.4 |
| COX2 | 81/94 | 41.2 | 43.5 |
| DD | 390/236 | 284.3 | 715.7 |
| NCI1 | 1646/822 | 29.8 | 32.3 |
| IMDB-B | 400/200 | 19.8 | 96.5 |
| REDDIT-B | 800/400 | 429.6 | 497.8 |

## F.2 Hyper-parameters

We select the key hyper-parameters of `SIGNET` through a group-level grid search. Specifically, the hyper-parameters for each benchmark dataset are demonstrated in Table 2. Note that for the dataset without ground-truth explanations, we would not tune the hyper-parameters for the subgraph extractor but use the default ones. The grid search is carried out on the following search space:

- Number of epochs $E$: {10, 50, 100, 200, 500, 800, 1000}

- Learning rate $lr$: {1e-2, 1e-3, 1e-4}

- Layer number of encoders $L_{enc}$: {2,3,4,5}

- Hidden dimension of encoders $D_{enc}$: {16,32,64,128}

- Model type of subgraph extractor $EXT$: {MLP,GIN}

- Layer number of subgraph extractor $L_{ext}$: {2,3,4,5}

- Hidden dimension of subgraph extractor $D_{ext}$: {4,8,16,32}

To ensure robust and reliable results, we also conducted a comprehensive grid search to obtain the best hyperparameter configurations for the baselines. Specifically, for deep GLAD methods (i.e., OCGIN, GLocalKD, and OCGTL), we performed grid searches on both general hyperparameters (e.g., layer number and hidden dimensions) and model-specific hyperparameters (e.g., the number of transformations in OCGTL). Similarly, for post-hoc explainers, we conducted grid searches on their post-hoc training iterations and learning rates. As for shallow GLAD methods, we focused on searching for key hyperparameters such as the training iterations of detectors and kernel-specific parameters.

## F.3 Metrics for explanation performance evaluation

We tackle the explanation problem by framing it as a binary classification task for nodes and edges. We designate nodes and edges inside the explanation subgraph as positive instances and the rest as negative. The importance weights generated by the explanation methods serve as prediction scores. An effective explanation method should assign higher weights to nodes and edges within the ground truth subgraphs compared to those outside. To quantitatively evaluate the performance, we use the AUC as the metric for this binary classification problem. A higher AUC indicates better performance in providing meaningful explanations.

Table 2: Details of the hyper-parameters tuned by grid search.

| Dataset | $E$ | $lr$ | $L_{enc}$ | $D_{enc}$ | $EXT$ | $L_{ext}$ | $D_{ext}$ |
|---|---|---|---|---|---|---|---|
| BM-MT | 1000 | 1e-2 | 5 | 16 | GNN | 2 | 16 |
| BM-MN | 500 | 1e-2 | 5 | 16 | GNN | 3 | 8 |
| BM-MS | 200 | 1e-2 | 5 | 16 | GNN | 2 | 32 |
| MNIST-0 | 50 | 1e-2 | 2 | 16 | MLP | 2 | 16 |
| MNIST-1 | 50 | 1e-2 | 2 | 16 | MLP | 2 | 16 |
| MUTAG | 50 | 1e-2 | 5 | 16 | GNN | 5 | 4 |
| PROTEINS-F | 800 | 1e-3 | 5 | 16 | GNN | 5 | 8 |
| ENZYMES | 1000 | 1e-3 | 5 | 128 | GNN | 5 | 8 |
| AIDS | 1000 | 1e-4 | 5 | 16 | GNN | 5 | 8 |
| DHFR | 1000 | 1e-4 | 5 | 128 | GNN | 5 | 8 |
| BZR | 1000 | 1e-4 | 5 | 128 | GNN | 5 | 8 |
| COX2 | 1000 | 1e-4 | 5 | 64 | GNN | 5 | 8 |
| DD | 100 | 1e-4 | 5 | 128 | GNN | 5 | 8 |
| NCI1 | 1000 | 1e-4 | 5 | 128 | GNN | 5 | 8 |
| IMDB-B | 10 | 1e-4 | 5 | 64 | GNN | 5 | 8 |
| REDDIT-B | 1000 | 1e-4 | 5 | 128 | GNN | 5 | 8 |

### F.4 Implementation of GLAD methods with post-hoc explainers

Given a GLAD model and post-hoc explainer, at first, we train the GLAD model independently on the training set. After sufficient training, the GLAD model is able to map each input graph into a scalar, i.e., its anomaly score. To address the uncertainty of the anomaly score boundaries, we apply a linear scaling function to map the scores into the [0,1] range and then use a sigmoid function to convert each score into a probability for binary classification. Subsequently, we integrate the post-hoc explainer with the probabilized output of the GLAD model and optimize the explainer accordingly.

### F.5 Computing infrastructures

We implement the proposed `SIGNET` with PyTorch 1.12.0 [60] and PyTorch Geometric (PyG) 2.3.0 [61]. The experiments are conducted on a Linux server with an Intel Xeon E-2288G CPU and two Quadro RTX 6000 GPUs.

## G  Further Supplementary of Qualitative Experiments

More visualization of explanation results by `SIGNET` are given in Fig. 2. In specific, we visualize the node-level and edge-level probabilities on four datasets, i.e., BM-MT, BM-MN, BM-MS, and MUTAG. For each dataset, the top row includes 5 normal examples, and the bottom row includes 5 anomalous examples. For MUTAG dataset, the normal examples do not have a specific rationale, while the rationales for anomalies are -NO2 or -NH2.

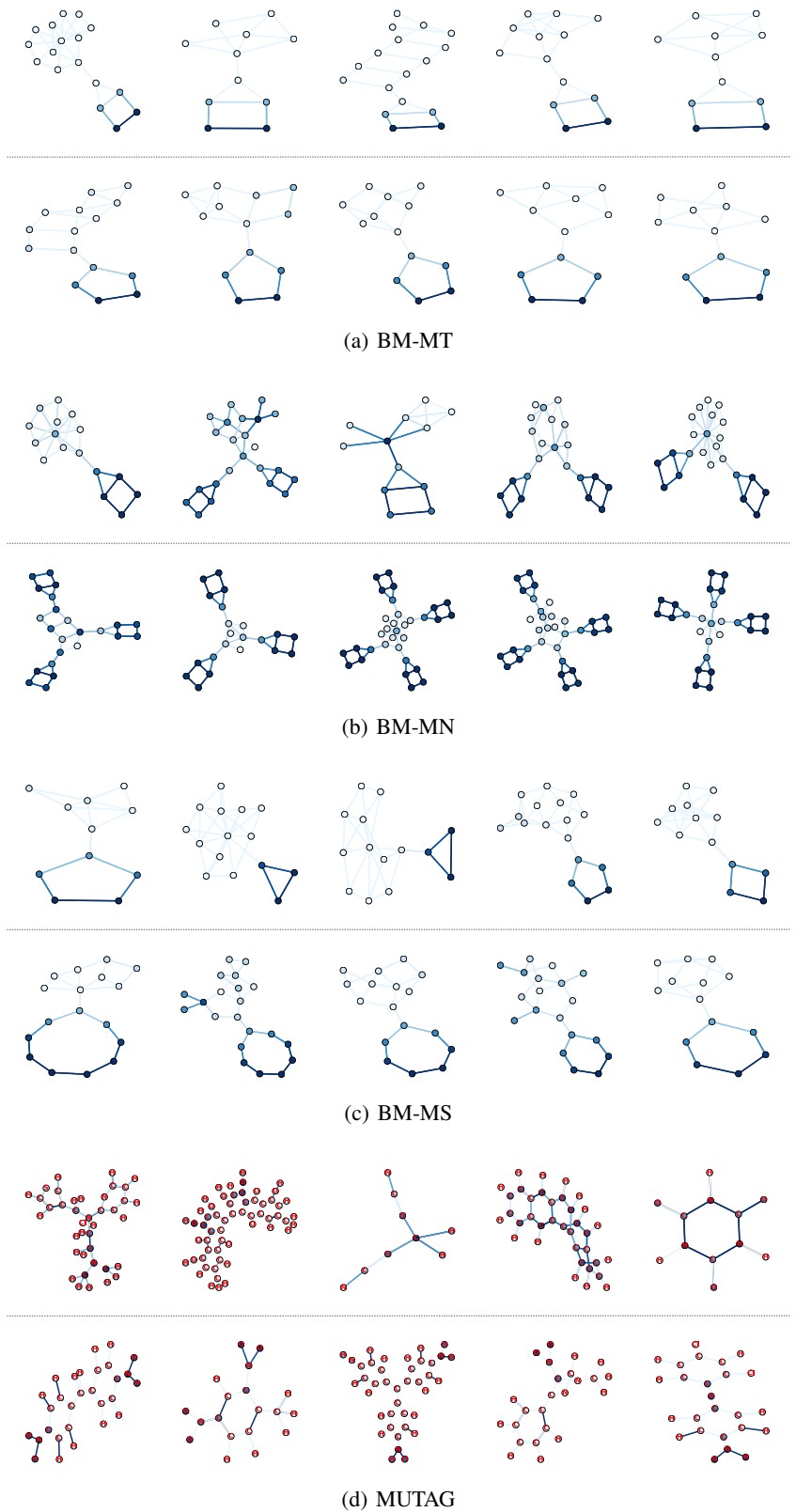

Figure 2: Visualization of explanation results w.r.t. node and edge probabilities. For each dataset, the top row includes 5 normal examples, and the bottom row includes 5 anomalous examples.