# OpenReview forum: "Towards Self-Interpretable Graph-Level Anomaly Detection"
_NeurIPS.cc/2023/Conference — NeurIPS 2023 poster_

### Official Review · Reviewer_QzdR · 2023-06-20

**Soundness:** 2 fair
**Presentation:** 3 good
**Contribution:** 2 fair
**Rating:** 5
**Confidence:** 3

**Summary:**

Inspired by the multi-view information bottleneck and dual hypergraph transformation, this paper proposed a self-explainable anomalous graph detection method called SIGNET. The method has three key designs in the model training stage: 1) Dual hypergraph-based view construction, 2) Bottleneck subgraph extraction, and 3) Cross-view maximization. Anomalous graph detection and graph rationale extraction will be achieved through maximizing the mutual information between the representations of graph rationale extracted from the original graph view and hyper-graph view. During inference, the anomaly score of a test graph is determined by the negative mutual information between the representations of its two graph rationales from different views.

**Strengths:**

1. The problem studied in this paper is important and interesting. It is essential to understand the subgraphs make a graph anomalous.
2. This paper provides a clear summary of the challenges of self-interpretable GLAD and why existing explainers for GNNs cannot be applied to graph-level anomaly detection.
3. This paper is well-structured.


**Weaknesses:**

1. Despite having a clear framework, it is still hard to get the motivations behind employing multi-view information bottleneck (MIB) and dual hypergraph transformation (DHT) to solve self-interpretable graph-level anomaly detection.  Much like GAST [1], authors draw inspiration from information theory to extract subgraphs as the explanation of model prediction, but they fail to adequately discuss the underlying intuition or theoretical basis for how MIB and DHT specifically address graph-level anomaly detection.
2. The dataset used in experiments is relatively small-scale, and time complexity analysis is lacking. If there are difficulties in conducting experiments on large-scale graph data, time complexity analysis is necessary.
3. More related work about anomaly detection and explainers of anomaly detection is expected.

[1] Miao et al. Interpretable and Generalizable Graph Learning via Stochastic Attention Mechanism, ICML 2022.



**Questions:**

1. How does the equation (1) improve unsupervised representation? In other words, why does the Mutual information between $V_1$ and $Z_1$ need to be minimized when $V_2$ is known? And, why does the mutual information between $V_2$ and $Z_1$ need to be maximize?
2. What are the key connections between unsupervised representation and graph-level anomaly detection? In this paper, graph-level anomaly detection is not a completely unsupervised problem, because labelled normal graphs work as the supervision information.
(The answers to the above two questions 1 and 2 will help me understand why MIB can be applied to graph-level anomaly detection problems.)
3. Why DHT-based view is useful for detecting anomalous graphs? More fine-grained discussions are important. For example, for a normal graph and an anomalous graph, what does their DHT-based view look like? If this is difficult to depict, it would be beneficial to explain why, for a normal graph, the mutual information between the representation of subgraph extracted from the original graph view and the DHT-based view should be high, while for an anomalous graph, the mutual information between representations of two different views should be low when two graph views share a single extractor?  (I have read the content from line 210 to line 215.  question: why the distinct content between dual hyper-graph and original view can benefit anomaly detection? What exactly are these distinctions?).
4. It is reasonable to use the same extractor to replace minimizing the SKL divergence?
5. Questions about the Loss function (5): Minimizing (5) is equal to maximizing $I(\mathbf h_{G_i^{(s)}}, \mathbf h_{G_i^{\star (s)}})$. Hence, for $l(\mathbf h_{G_i^{(s)}}, \mathbf h_{G_i^{\star (s)}})$, a larger numerator and a smaller denominator are expected. My question is, what is the meaning of the small denominator? Why do we need to make the representation $\mathbf h_{G_i^{(s)}}$ of graph rationales about normal graph $i$ to be dissimilar from $\mathbf h_{G_j^{(s)}}$ representation of graph rationales about normal graph $j$? Shouldn't normal graphs have similar representations (in experiments, authors use the graphs from the same class as normal graph)? In this way, the representations of their graph rationales should also be similar.
6. In experiments, how to use GE and PE to explain the predictions of OCGIN, GLocalKD, and OCGTL? The loss function of GE and PE is designed for graph classification.
7. The proposed method has connection with infoNCE loss and contrastive learning, a contrastive learning-based graph-level anomaly detection baseline is needed. GOOD-D [2] is a suitable choice.

[2] Liu et al, GOOD-D: On Unsupervised Graph Out-Of-Distribution Detection, WSDM 2023.



**Limitations:**

yes

---

> ### Author Rebuttal · Authors · 2023-08-10
>
> **Reply to Reviewer QzdR**
>
> We are grateful to Reviewer QzdR for providing insightful feedback. Due to the limitation of space, we only reply to some core questions as below. For Questions 1/4/6 in the review, we will discuss them later.
>
> **Q1: Motivation behind using MIB and DHT**
>
> **A1:** We appreciate the reviewer for the insightful comment. We discuss the motivations as follows:
>
> **MIB**: We want to mention that in (unsupervised) graph-level anomaly detection (GLAD), the training phase only contains normal graphs, and the goal is to detect abnormal graphs during the test phase. Since we do not have any knowledge of labeled abnormal graphs during training, it is hard to learn a precise decision boundary between normal and anomalous graphs. Due to the lack of ground-truth knowledge of abnormal graphs during training, we try to leverage self-supervised (unsupervised) learning, in particular MIB to build the GLAD model, which characterizes the distribution of normal graphs and provides a good indicator for measuring the abnormality of test graphs. Also, by extracting the bottleneck subgraphs from our MSIB framework, we are able to provide self-interpretability when detecting anomalies.
>
> **DHT**: As our goal is to learn the normal data distribution through self-supervised learning, however, simply augmenting the input graph with stochastic perturbation may introduce anomalous graphs, leading to unexpected performance degradation. In contrast, the DHT-based view construction avoids perturbing the original graph's semantic knowledge and provides a new view of the input graph, enabling us to capture consistent matching patterns between the two views. Additionally, the dual hypergraph approach prioritizes edge-level information, encouraging the model to recognize both node-level and edge-level anomaly patterns. Consequently, the node-edge matching pattern becomes a focal point for model learning and serves as an indicator for normal/abnormal graphs. For instance, if a normal pattern is a star-shaped motif, the node patterns (e.g., high-degree center node) and edge patterns (e.g., connections between center/tail nodes) of this motif can be effectively captured by maximizing the agreement between the original view and the DHT view. In cases where an anomaly with a similar but distinct motif (e.g., clique) is introduced, it would exhibit a lower mutual information (MI) value due to the disruption of node-edge matching patterns, which can help us to better detect the abnormal graphs during the test phase.
>
> **Q2: Experiments on large-scale datasets and complexity analysis**
>
> **A2:** Thank you for your valuable feedback regarding the scalability and time complexity analysis. In response to your suggestions, we conducted additional experiments on two large-scale datasets with more samples, BM-MT-20k and MNIST-0-6k. The content of datasets is similar to BM-MT and MNIST-0. The results below demonstrate that our method can handle datasets with a large number of graphs, verifying the scalability of our method. Note that SIGNET can also handle datasets with large graphs (e.g., REDDIT-B with 429.6 avg. #node), as shown in our main paper. Furthermore, we have included a comprehensive time complexity analysis of our method, which is attached in P1 of general response.
>
> |Dataset|AD-AUC|NX-AUC|EX-AUC|Runtime|
> |-|-|-|-|-|
> |BM-MT-20k|95.75%|81.85%|80.59%|2.5561s|
> |MNIST-0-6k|82.25%|74.08%|76.35%|0.6247s|
>
> **Q3: Connections between unsupervised representation and graph-level anomaly detection**
>
> **A3:** We appreciate the reviewer for raising this concern. Following the previous literature on GLAD, we want to mention that the training graphs only contain normal ones, and the goal is to detect abnormal graphs during the test phase. Since we do not have any knowledge of labeled abnormal graphs during training, it is hard to learn precise decision boundaries between normal and anomalous graphs. Therefore, we try to leverage unsupervised/self-supervised learning to build the graph-level anomaly detection model, which characterizes the distribution of normal graphs and provides an indicator (i.e., cross-view MI) for measuring the abnormality of test graphs.
>
> **Q4: Why minimize the MI between graph rationale and other normal graphs**
>
> **A4:** We appreciate your insightful question. Firstly, the Info-NCE mutual information estimator theoretically requires negative samples in its denominator to prevent collapse, i.e.,  all input pairs are estimated to have large MI. To meet this requirement efficiently, we opted to use other samples in the same batch as negative samples, which is an efficient strategy without extra cost. During the model design phase, we explored stochastic graph perturbation to generate synthetic negative samples. However, this approach proved less effective, unstable, and time-consuming. Empirically, using other normal samples as negative samples has shown effectiveness. Secondly, despite normal samples belonging to the same class or having similar properties, their graph rationales can differ from each other. For instance, in the BM-MS dataset, the rationales may include rings of different sizes; in the MUTAG dataset, they can be -NH2 or -NO2. Hence, maximizing the similarity between a graph and the rationales of other graphs may lead to mismatching and hurt stability and performance. Instead, maximizing the similarity of each graph to its own rationale proves to be a reliable strategy. Therefore, we opted to use Info-NCE to ensure that each graph has a larger similarity to its own rationale rather than to others' rationales. We hope this explanation addresses your query and provides better insight into our approach.
>
> **Q5: New baseline**
>
> **A5:** Thank you for providing a strong anomaly detection baseline GOOD-D that makes our experiments more convincing. We will include a comparison with GOOD-D, see attached PDF, where SIGNET outperforms GOOD-D with a significant performance gap of 7.40%.

---

> > ### Author Response · Authors · 2023-08-10
> > **Discussion for more questions**
> >
> > We extend our gratitude to Reviewer QzdR for their insightful feedback. In this section, we offer responses to the remaining questions posed by the reviewers. If you have any further points for discussion, please feel free to share them here.
> >
> > **Q6: Regarding the explanation of MIB principle (Eq. (1))**
> >
> > **A6:** We appreciate the reviewer for the thoughtful question. Our explanation of Eq. (1) is mainly based on the original paper of multi-view information bottleneck (MIB) where this equation is deducted [*1]. According to the paper, minimizing the mutual information between $V_1$ and $Z_1$ conditioned on $V_2$ can improve unsupervised representation by discarding irrelevant information that is unique to $V_1$ and not predictable by observing $V_2$. This is because, in the multi-view unsupervised setting, we assume mutual redundancy between the two views, meaning that they share some information. Therefore, any information that is unique to one view and not shared by the other is considered superfluous and can be safely discarded without losing any label information. On the other hand, maximizing the mutual information between $V_2$ and $Z_1$ is necessary to ensure that the representation is sufficient for the potential label $Y$. This is because the mutual information between $V_2$ and $Z_1$ represents how much label information is accessible from the representation. Therefore, by maximizing this term, the authors are able to ensure that the representation contains enough information to predict the label accurately. Overall, by combining these two requirements using a relaxed Lagrangian objective, we are able to obtain a minimal sufficient representation that discards as much superfluous information as possible without losing any label information, resulting in a more robust and informative representation. In the revised paper, we will discuss more about the inner mechanism of MIB.
> >
> > [*1] Federici et al. Learning robust representations via multi-view information bottleneck. In ICLR, 2020.
> >
> > **Q7: Is it reasonable to use the same extractor to replace minimizing the SKL divergence?**
> >
> > **A7:** Thank you for your insightful question, and our answer is indeed positive. Theoretically, the purpose of the SKL divergence term is to align two distributions, specifically $p(G^{1(s)}|G^{1})$ and $p(G^{2(s)}|G^{2})$. When these distributions are perfectly matched, the SKL value diminishes to $0$. It's important to note that the transformations from $G$ to both $G^{1}$ and $G^{2}$ are deterministic. If we employ the same extractor to generate $G^{1(s)}$ and $G^{2(s)}$ from $G=G^{1}$, this implies an inherent alignment of the two distributions, resulting in a $D_{SKL}$ of $0$, which aligns with our intention. Hence, we can omit the SKL term from the objective function. Furthermore, the empirical comparison in Section 4.4 underscores the effectiveness of this design. To conclude, based on both theoretical consideration and experimental evidence, we assert that this design choice is reasonable.
> >
> > **Q8: Details of “detector+explainer” baselines**
> >
> > **A8:** We appreciate your insightful question. Firstly, the Info-NCE mutual information estimator theoretically requires negative samples in its denominator to prevent collapse [*2,*3], i.e.,  all input pairs are estimated to have large MI. To meet this requirement efficiently, we opted to use other samples in the same batch as negative samples, which is an efficient strategy without extra cost. During the model design phase, we explored stochastic graph perturbation to generate synthetic negative samples. However, this approach proved less effective, unstable, and time-consuming. Empirically, using other normal samples as negative samples has shown effectiveness. Secondly, despite normal samples belonging to the same class or having similar properties, their graph rationales can differ from each other. For instance, in the BM-MS dataset, the rationales may include rings of different sizes; in the MUTAG dataset, they can be -NH2 or -NO2. Hence, maximizing the similarity between a graph and the rationales of other graphs may lead to mismatching and hurt stability and performance. Instead, maximizing the similarity of each graph to its own rationale proves to be a reliable strategy. Therefore, we opted to use Info-NCE to ensure that each graph has a larger similarity to its own rationale rather than to others' rationales. We hope this explanation addresses your query and provides better insight into our approach.
> >
> > [*2] Tschannen et al. "On Mutual Information Maximization for Representation Learning." In ICLR, 2020.
> >
> > [*3] Chen et al. "A simple framework for contrastive learning of visual representations." In ICML, 2020.
> >
> > **Q9: More related works**
> >
> > **A9:** We appreciate the reviewer for the suggestion of our literature review. In the revised version of the paper, we will conduct a thorough review and include more references on both anomaly detection and explainable anomaly detection.

---

> > > ### Comment · Reviewer_QzdR · 2023-08-16
> > >
> > > Thanks for the reply. I have raised the rating to 5.

---

> > > > ### Author Response · Authors · 2023-08-17
> > > > **Thanks for the raising**
> > > >
> > > > We sincerely thank Reviewer QzdR for proposing this enhancement! If you have any further questions or concerns, we are willing to engage in further discussions.

---

### Official Review · Reviewer_vtQs · 2023-06-27

**Soundness:** 3 good
**Presentation:** 4 excellent
**Contribution:** 3 good
**Rating:** 6
**Confidence:** 5

**Summary:**

The paper tackles the task of graph anomaly detection. It proposes a multi-view subgraph information bottleneck framework that is used to introduce SIGNET, a self-interpretable graph anomaly detection method. The method leverages the dual hypergraph transformation to obtain two views of the same graph, which are subsequently used for a mutual information maximization objective. The resulting model can then be employed to detect anomalous samples in the dataset and provides an implicit explainability mechanism that can highlight salient regions of the graph.


**Strengths:**

- The paper is well-written, and the ideas presented are exposed in an easy-to-follow manner.

- The authors present compelling arguments for their design choices, an ablation study, and the derivation of the loss function in the Supplementary materials.

- The Supplementary materials contain the code for the proposed approach and various details about the experimental setup, such as the hyperparameter pool used during hyperparameter selection.

- The quantitative results are compelling, and the proposed method obtains the best results in most cases. Adequate baselines were selected for the experiments, containing both neural approaches and more classical AD methods, such as the OC-SVM with a WL kernel.

**Weaknesses:**

- The paper focuses on presenting an explainability/interpretability mechanism instead of an anomaly detection pipeline with an explainability mechanism built in. The usage of the dual hypergraph view in the context of training an anomaly detector is novel in its own right. The main results for anomaly detection in Table 2 are also good. An explainability mechanism is a very important addition, but I would prefer it not to be the paper's primary focus. The strong emphasis on explainability made me think that the authors were not confident in the anomaly detection model, which in my eyes, should be pitched as the main strength of the paper.

- The qualitative analysis of the explanations is somewhat lacking. A more detailed caption for Fig. 3, which also explains what the highlighted nodes and edges represent, would improve readability. It looks to me that the model highlights the motifs in both the normal and anomaly scenarios, but I would have expected it to highlight the motif in the anomalous graphs more. I would also like more qualitative results, especially on real-world datasets. Figure 3 is overall somewhat confusing and makes me feel like the examples might be cherry-picked.

- As far as I'm aware, the NX-AUC and EX-AUC metrics are not very common. Please consider expanding their meaning in the main paper or the Supplementary materials.

- I don't particularly like the "cartoon"-style fonts used in Fig. 1 and Fig. 2., please consider some alternatives. However, this is a subjective stylistic opinion and won't affect my final score for the paper.

- I have not seen any mention of hyperparameter selection for the baselines. Some classical methods (such as the OC-SVM) are particularly sensitive to hyperparameters. Please consider adding a discussion regarding this.

- It would be insightful if the paper would further discuss potential limitations and negative results.

**Questions:**

- Do the datasets used contain anomalies in the training set? I would love to see an analysis of the performance with respect to the anomaly contamination percentage in the training data.

- The overall objective would also be interesting for general unsupervised graph representation learning. Have the authors tried using the graph representations for downstream tasks? I'm not requesting this experiment to be done since it's outside the scope of the paper; it's just something I have thought about while reading it. I imagine the results would not be great since the two views are somewhat similar.

- Did the authors search for any hyperparameters for the baselines?

- Did the authors examine any other qualitative results?

**Limitations:**

- The authors have addressed their limitations regarding the inability to use ground-truth labels when training their model, making it entirely unsupervised. Expanding the discussion of the limitations, potentially with negative qualitative results, would be beneficial. Discussing FP/FN samples obtained by picking some anomaly threshold would also be interesting since the model could provide explanations.

---

> ### Author Rebuttal · Authors · 2023-08-10
>
> **Reply to Reviewer vtQs**
>
> We appreciate Reviewer vtQs for the valuable feedback and acknowledge our technical contributions and the effectiveness of the proposed method. We address the concerns raised by the reviewer as follows.
>
> **Q1: The main focus of this paper**
>
> **A1:** Thank you for your valuable feedback and perspective on our paper. We want to clarify that the primary focus of our paper is to develop a self-interpretable graph-level anomaly detection method with robust anomaly detection capability and promising explainability for its predictions. We are confident in both of these capabilities, and they have been substantiated by the results of our experiments in Sections 4.2 and 4.3, respectively. However, it should be noted that self-interpretability is a novel property for graph-level detection methods, representing a "zero-to-one" innovation. As a result, we have placed additional emphasis on introducing and explaining this innovative aspect, which may inadvertently overshadow the strength of our anomaly detection model. We appreciate your valuable insight, and based on your suggestion, we will revise our paper to ensure a more balanced presentation of both the detection capability and the novel self-interpretability feature.
>
> **Q2: Insufficient qualitative analysis**
>
> **A2:** Thanks for the valuable suggestion! We have prepared more visualization results, which are included in the attached PDF and will be incorporated into the revised version of the paper. These additional visualizations cover not only three synthetic datasets but also a real-world dataset, namely MUTAG. To address your concern about the confusion in Figure 3, we will provide a more detailed caption that explains what the highlighted nodes and edges represent. Also, we will discuss the new results in more depth. Notably, the visualization examples are randomly selected rather than cherry-picked. As a result, you can witness some cases where SIGNET does not work perfectly in our new results.
>
> **Q3: The meaning of evaluation metrics**
>
> **A3:** We understand the importance of clarity and ensuring that the metrics used in our study are well-defined for readers. In the revised version of the paper, we will provide a more detailed explanation (P3 in general response) of the NX-AUC and EX-AUC metrics.
>
> **Q4: Hyper-parameters of baselines**
>
> **A4:** We apologize for the oversight in not mentioning the hyperparameter selection process for the baselines. Allow us to provide more clarity on this matter. To ensure robust and reliable results, we conducted a comprehensive grid search to obtain the best hyperparameter configurations for the baselines. Specifically, for deep GLAD methods, we performed grid searches on key hyperparameters (e.g., layer number and hidden dimensions). For post-hoc explainers, we conducted grid searches on their post-hoc training iterations and learning rates. As for shallow GLAD methods, we focused on searching for key hyperparameters such as the training iterations of detectors and kernel-specific parameters. We are committed to adding a more specific discussion regarding this in the revised version of the paper.
>
> **Q5: More discussion about limitations**
>
> **A5:** We appreciate your suggestion to discuss potential limitations and negative results in our paper.  In the revised version of our paper, we will include a dedicated section to address the potential limitations of our proposed method, SIGNET, and also discuss any negative results that we encountered during our research. For the method, While SIGNET performs well across various datasets, there might be specific datasets where its performance is not optimal. We will discuss the dataset characteristics that could impact the method's generalization and suggest ways to address this issue. For the negative results, we will transparently share any experiments or scenarios where SIGNET did not perform as expected or where its interpretability might be limited.
>
> **Q6: Anomalies in training set**
>
> **A6:** Thanks for the thoughtful comment! In this study, we adhere to the common practice of unsupervised graph-level anomaly detection, using a training set comprising only normal samples without any anomalies. Theoretically, adding anomalies to the training set may decrease the performance of all methods, including our proposed approach. However, it is important to note that some baseline methods are not specifically designed to handle scenarios with training data containing anomalies. Consequently, including anomalies in the training set could lead to unfair comparisons, as these baseline methods might not perform optimally in such settings. Although we recognize the significance of analyzing performance with varying levels of anomaly contamination in the training data, we have decided to leave this investigation for future work.
>
> **Q7: Style of figures**
>
> **A7:** Thanks for the valuable suggestion. In our figures, the cartoon-style fonts were chosen to provide a high-level explanation of the research question and the proposed method. The intention was to present a clear and visually engaging representation of the high-level concepts in another style, without delving into the specifics of the learning process. However, we also recognize the importance of maintaining a balance between clarity and aesthetics. In the revised version, we will explore alternative font styles that can still convey the high-level explanation effectively while aligning with academic standards.
>
> **Q8: Potential applications on unsupervised graph representation learning**
>
> **A8:** We agree that exploring the use of our framework for unsupervised graph representation learning could be promising. However, in this paper, our main focus is on the specific problem of unsupervised graph-level anomaly detection. In future works, we will consider exploring the use of our framework for unsupervised graph representation learning and its impact on various tasks.

---

> > ### Comment · Reviewer_vtQs · 2023-08-16
> >
> > I thank the authors for their clarifications to me and the other reviewers.
> >
> > I will keep my score of 6 and raise my confidence to 5.

---

> > > ### Author Response · Authors · 2023-08-17
> > > **Thanks for the reply and comments**
> > >
> > > We sincerely appreciate the reviewer once again for recognizing our contribution and for providing valuable comments! Your insights are truly invaluable to us.

---

### Official Review · Reviewer_yuBa · 2023-06-29

**Soundness:** 3 good
**Presentation:** 3 good
**Contribution:** 2 fair
**Rating:** 5
**Confidence:** 4

**Summary:**

To overcome the disadvantage of existing anomaly detection methods that fail to provide meaningful explanations for the predictions, this paper proposes SIGNET to (1) detects anomalous graphs and (2) generate informative explanations. To achieve this, the paper devises a multi-view subgraph information bottleneck framework to extract the informative subgraphs as explanations. Empirical results on 16 datasets verify the effectiveness of the proposed method.

**Strengths:**

- The investigated problem is novel and important, as robustness and interpretability are the key sub-areas of trustworthy graph learning.
- The paper is easy to follow, with generally clear writing and illustration.
- The proposed SIGNET is technically solid, with good and competitive empirical results.
- Some preliminary analyses in terms of information theory are conducted.

**Weaknesses:**

- The technical contributions are neutral. The proposed MSIB framework seems to be a combination of information bottleneck and graph multi-view learning, while the novelty and difficulty are not clear.
- The answers to "RQ1: Can SIGNET provide informative explanations for the detection results" are not convincing enough, which should be the key contribution of the paper. The reasons are as follows.
- The two compared GNN explainers, GNNExplainer and PGExplainer, are not up-to-date. The latest and state-of-the-art explainers, e.g., GSAT (ICML'22) [22], should also be considered and discussed.
- Note that GSAT is also derived from the information bottleneck, which shares a similar design as the proposed SIGNET. I would suggest the paper discuss more the connection and differences.
- Besides, directly combining GNN explainers and anomaly detection methods, e.g., OCGIN-GE, can be sub-optimal, as shown in Table 1. The paper should explain more about the baseline settings, that is, why such a combination is reasonable, but the results show that it does not work well in most cases.
- The few cases shown in Figure 3 are insufficient, which are not promising and convincing enough. It seems that SIGNET learns to capture the same (similar) functional sub-graph (or motif) for both normal and anomaly samples. I would suggest the paper show more cases and provide an in-depth analysis, which will add more value to the main contribution, i.e., interpretability.

**Questions:**

- For the anomaly detection performance, Table 2 shows that SIGNET is outperformed by the three baselines in some cases. What are the reasons here? Is there a natural tradeoff between accuracy and interpretability?
- The MI estimation appears many times. How does the paper conduct the MI estimation in a tractable and differentiable way? Equation (3) seems confusing for including the MI and SKL divergence.
- Besides, how does Equation (5) correlate with Equation (3), is Equation (5) also an MI estimation?

**Limitations:**

Please refer to the above Weaknesses and Questions.

I would consider raising my score if the above questions are well answered.

---

> ### Author Rebuttal · Authors · 2023-08-10
>
> **Reply to Reviewer yuBa**
>
> We appreciate Reviewer yuBa for the perception of our contributions and thank the reviewer for the insightful feedback. We provide our responses as follows.
>
> **Q1: Technical contribution of this paper**
>
> **A1:** Thanks for the valuable feedback. Our work embarks on an important and challenging research direction, representing the first step towards integrating self-interpretability into graph-level anomaly detection (GLAD), thereby serving as an inspirational catalyst for future advancements in this domain. It's essential to note that our proposed method transcends a mere combination of existing techniques. The Multi-View Subgraph Information Bottleneck (MSIB) framework is thoughtfully crafted to address the demands of unsupervised self-interpretable GLAD with sound motivation and solid deduction. Building upon this framework, we introduce several innovative designs to effectively tackle the intricacies of the research problem. We greatly value the reviewer's feedback and wish to emphasize that our paper introduces a pioneering GLAD method with inherent self-interpretability, carving a unique path distinct from existing approaches, and our work holds the potential to inspire future research in the evolving field of GLAD.
>
> **Q2: Compared to state-of-the-art explainers**
>
> **A2:** Thanks for the kind suggestion. We understand the importance of considering other state-of-the-art explainers for comparison. However, regarding GSAT, we acknowledge that it is a self-interpretable GNN that requires labels for training. Unfortunately, in our GLAD setting, we do not have access to labels. As a result, we cannot directly compare our approach with GSAT.
>
> To address the reviewer's concern and ensure comprehensive evaluation, we have introduced one of the latest post-hoc explainers, RC-Explainer (TPAMI 2022 [1]), for comparison. The comparison (w.r.t. EX-AUC) below shows that SIGNET continues to outperform RC-Explainer on three datasets, reaffirming that the combination of trained detectors and post-hoc explainers typically provides sub-optimal explainability.
>
> |Methods|BM-MN|MNIST-0|MNIST-1|
> |-|-|-|-|
> |GLocalKD-RC|71.87|63.56|61.67|
> |OCGTL-RC|68.50|66.83|63.22|
> |SIGNET|83.45|72.78|74.83|
>
> [1] Wang et al. "Reinforced causal explainer for graph neural networks." IEEE TPAMI (2022).
>
> **Q3: Connection and differences with GSAT**
>
> **A3:** Thanks for the valuable comment. Following the reviewer's suggestion, we will include the connections and differences between our method (SIGNET) and GSAT in the revised paper, see P4 in general response.
>
> **Q4: Discussion about “detector+explainer” baselines**
>
> **A4:** Thanks for your valuable feedback! Regarding the implementation details, we will add them to the revised paper and we also attach them to P2 in general response. We claim that the combination is reasonable because the mechanism of post-hoc explainers is to parameterize the input graph and find the input that can generate the ideal output under several conditions, serving as the explanation. Since the baseline GLAD methods are deep learning models with graph-level input and scalar output, it is feasible to use the post-hoc explainer to explain their prediction, similar to the explanation process of classification models.
>
> However, such a combination would lead to sub-optimal performance since the GLAD model and explainer are trained independently, which might not ensure optimal alignment between the two components during training. Consequently, this leads to the mismatch between the generated explanations and true decision boundaries learned by GLAD model. In contrast, SIGNET avoids this potential mismatch by incorporating interpretability into the detection model, allowing for joint learning of prediction and interpretation. By optimizing a unified objective for detection and explanation, SIGNET can align its learned decision boundaries with the generated explanations effectively.
>
> **Q5: More case studies**
>
> **A5:** Thanks for the valuable suggestion! In response to your feedback, we have prepared more visualization results (including real-world dataset MUTAG), which are included in the attached PDF. With the new results, we will provide a detailed analysis in the revised paper.
>
> **Q6: Discussion about anomaly detection (AD) performance**
>
> **A6:** Thanks for the kind suggestion. While SIGNET has demonstrated strong AD performance on most benchmark datasets, we acknowledge that there are cases where it is outperformed by some of the baselines. A possible reason for this is the dataset characteristics. Different datasets may have varying levels of complexity and distribution of anomalies, which can impact the performance of AD methods. Some baselines might be more suitable for certain datasets due to their specific design and assumptions, leading to their better performance.
>
> **Q7: Details of MI estimation in SIGNET**
>
> **A7:** Thanks for the thoughtful comment. MI plays a critical role in our proposed MSIB framework, specifically in the first term of Eq.(3). However, estimating MI can be challenging, especially with limited examples instead of the variable distribution itself. Fortunately, [2] introduced parametrized MI estimators that offer a tractable and differentiable way to estimate MI using neural networks. In SIGNET, we adopt the Info-NCE (Eq.(5)) for MI estimation due to its generalization ability (see the 2nd paragraph of Sec. 3.4). Experiments in Sec. 4.4 shows the superiority of Info-NCE over other estimators.
>
> To correlate Eq.(3) with Eq.(5), instead of minimizing the SKL between two extractors, we utilize a unified subgraph extractor to estimate the subgraph for both views (see the 2nd paragraph of Sec. 3.3). This naturally aligns the two distributions without the need to minimize their SKL. Consequently, we can omit the second term in Eq.(3), leading to Eq.(5).
>
> [2] Tschannen et al. "On Mutual Information Maximization for Representation Learning." ICLR (2020).

---

> > ### Comment · Reviewer_yuBa · 2023-08-16
> >
> > Thanks for the clarification and extra results. Most of my concerns are alleviated. Nonetheless, the original novelty and technical contribution of the proposed method are neutral to me. Therefore, I will keep my score as 5 and raise my confidence to 4.

---

> > > ### Author Response · Authors · 2023-08-17
> > > **Appreciation for your response and inquiry regarding additional questions**
> > >
> > > We appreciate Reviewer yuBa for your valuable comments! We would like to inquire whether you have any remaining concerns at this point. If there are any previous questions that haven't been thoroughly addressed due to the limitations of the rebuttal space, please feel free to highlight them. Your feedback is instrumental in refining our work, and we are willing to engage in further discussion to provide clarity on any unclear points or concerns you might have.

---

### Official Review · Reviewer_6qZe · 2023-07-06

**Soundness:** 3 good
**Presentation:** 4 excellent
**Contribution:** 3 good
**Rating:** 6
**Confidence:** 4

**Summary:**

This paper studies graph-level anomaly detection (GLAD), which aims to find the anomalous graphs. To construct an explainable GLAD under an unsupervised manner, the authors first proposed a multi-view subgraph information bottleneck (MSIB) framework and then introduce a dual hypergraph as a supplemental view of the original graph. The core contribution of the paper is the designing of a self-explainable GLAD model.

**Strengths:**

1. Explainable AI is important for many real-world applications that highlight interpretability and security. The proposed framework is explainable by the model itself (rather than post-hoc explainers).

1. Using two different and distinguishable views to train the MSIB framework for graph-level anomaly detection is reasonable and sound. The authors also used cross-view MI maximization for estimating MI between two compact subgraph representations.

**Weaknesses:**

1. The authors stated "this is the first attempt to study the explainability problem for anomaly detection on graph-structured data". However, there are works on node-level graph anomaly detection, both supervised and unsupervised and self-supervised. Better use "graph-level anomaly detection" here.

1. The proposed model is a little heavy and introduces non-trivial computational overhead. The trade-off between scalability and interpretability (there are also other efficient methods for explainable graph-level representation learning) should be considered. Complexity and run-time analysis should be reflected in the paper -- given the overhead of computing on multiple subgraphs.

1. Statistics of all datasets should be provided, e.g., avg. number of nodes, density of the graph.

1. All methods on GLAD seem to be unstable and less robust on the detection performance (with extremely high std). The evaluation of the effectiveness of the proposed method (RQ2 and RQ3) seems to be trivial and "boring". I suggest the authors to emphasize more on the RQ1.


**Questions:**

1. Suggestion: the authors should discuss more and provide more examples about the explanabilty of the model in RQ1 and in Figure 3. There are only visualizations for synthetic datasets. I believe readers would be more interested in these (rather than performance). The advantages of the proposed explanable GLAD model over existing explanable models should be clearly presented.

**Limitations:**

Authors slightly discussed the model limitations in the Conclusion section. More discussions w.r.t. complexity and dataset-type applicability are needed.

---

> ### Author Rebuttal · Authors · 2023-08-10
>
> **Reply to Reviewer 6qZe**
>
> We appreciate Reviewer 6qZe for the positive review and constructive comments. We provide our responses as follows.
>
> **Q1: Complexity and run-time analysis of the proposed method**
>
> **A1:** We appreciate your concern regarding the computational overhead and the trade-off between scalability and interpretability in our proposed model. In response to these concerns, we conduct a comprehensive time complexity analysis, which is attached to the general response and will display in the revised paper. The analysis shows that the time complexity of SIGNET is $\mathcal{O}(NLd^2(m+n) + Nnd(d_f+d_{f*}) + NBd)$, which is comparable to mainstream GNN-based models, including our baselines. From the analysis, we can also find that although the self-interpretable block (i.e., subgraph extractor) may introduce some additional time complexity, it is designed as a lightweight module compared to the entire anomaly detection framework and would not increase the overall scale of time complexity.
>
> Additionally, we provide a comparison between our method and baseline methods (including GOOD-D [1], the strong baseline pointed out by Reviewer QzdR) in terms of running time per epoch, as shown in the table below. This comparison illustrates that while our method provides self-interpretation capabilities, it still maintains competitive running efficiency. Specifically, on the dataset with larger graphs (MNIST-0), the runtime per epoch is very close to the most efficient baseline OCGIN, and is 6.5x faster than the strong baseline OCGTL. Therefore, the running efficiency of SIGNET would not be a large concern.
>
> |Dataset|OCGIN|GLocalKD|OCGTL|GOOD-D|SIGNET(ours)|
> |-|-|-|-|-|-|
> |BM-MT|0.0457s|0.0535s|0.2624s|0.3036s|0.0720s|
> |MNIST-0|0.1213s|0.3498s |0.8019s|0.5937s|0.1273s |
>
> [1] Yixin Liu, Kaize Ding, Huan Liu, and Shirui Pan. Good-d: On unsupervised graph out-of-distribution detection. In Proceedings of the Sixteenth ACM International Conference on Web Search and Data Mining, pages 339–347, 2023.
>
> **Q2: More discussion for explainability (RQ1)**
>
> **A2:** We appreciate your interest in the explainability of our proposed GLAD method and the importance of providing more examples and discussions. In the revised version of our paper, we will dedicate more attention to the model's explainability, discussing it in greater detail and providing additional examples. We acknowledge the significance of visualizations, as they offer a clear and intuitive understanding of the model's explanations. According to your suggestion, we have prepared more visualization results, which are included in the attached PDF and will be incorporated into the revised paper. These visualizations include the results on a real-world dataset MUTAG dataset. More visualization examples will be displayed when we open-source our code. I hope the extra results can address your concerns about the explainability of our method.
>
> **Q3: Detailed statistics of datasets**
>
> **A3:** Thanks for the kind suggestion! In response to the space limitations, we will move the dataset statistics to Appendix F in the supplementary material. We understand the importance of easy access, so we will clearly highlight the specific section in the revised version. Additionally, we will include more details, such as the density of graphs, as suggested by the reviewers.
>
> **Q4: Statement about “first attempt” contribution**
>
> **A4:** We appreciate your suggestion. While there have been previous works on node-level graph anomaly detection, our paper specifically focuses on the explainability problem for graph-level anomaly detection. To clarify this, we will use the term “graph-level anomaly detection” as our first-attempt contribution in the revised version. Thank you for helping us improve the accuracy of our paper.

---

> > ### Comment · Reviewer_6qZe · 2023-08-11
> > **Additional Questions**
> >
> > Thank you for the clarifications provided. Upon reviewing other questions, I have encountered further inquiries and hope that the authors could address my concerns. Before presenting my questions, I'd like to clarify that I possess a strong familiarity with hypergraphs, IB techniques, and self-supervised learning, as well as a reasonable understanding of (/un/semi)supervised time series anomaly detection. However, I must admit that I am not familiar with GLAD, including challenges, baselines, datasets, metrics, etc.
> >
> > - In light of a recent survey paper titled "A Survey on Explainable Anomaly Detection," it appears that the authors may not be the pioneers in considering explainability/interpretability in graph anomaly detection. Consequently, the assertions of being the "first attempt" or achieving a "zero-to-one" accomplishment might not be tenable. I agree with the first suggestion made by Reviewer vtQs in the weaknesses section. If the authors insist on emphasizing that the principal contribution of their work as the introduction of explainability/interpretability into GLAD, then the originality and significance of this work could be significantly weakened. My perspective is that the authors essentially utilize an existing IB technique to endow the model with a degree of interpretability, which, I must say, is a relatively limited advance. It's worth noting that IB has already found success in various similar tasks, such as graph/node/link classifications, to enhance explainability. I believe these tasks bear a strong connection to graph anomaly detection, especially considering that some of the datasets employed in this work were initially utilized in graph/node/link classification tasks.
> >
> > - I noticed the authors' description that they evaluated the anomaly detection performance of SIGNET using 10 TU datasets following the setting in [3]. After reading (or 'glancing through') [3], I have observed a discrepancy between the datasets utilized in this study (10 datasets) and those specified in [3] (16 datasets). Given that [3] is also compared as a strong baseline, it raises the question of why the experiments weren't conducted using precisely the same settings. Additionally, I am curious about the rationale behind using AD-AUC rather than the commonly employed AUC metric for evaluating model performance. Could the authors elaborate on the distinctions between these two metrics? While I understand that NX-AUC and EX-AUC metrics are chosen to assess explainability, I believe it is equally important for the authors to furnish AUC results using the same settings as the baselines.
> >
> > - As recommended by Reviewer QzdR, I echo the suggestion of providing a more comprehensive coverage of related works, particularly in the domain of explainable anomaly detection. Could authors offer a brief explanation here to illustrate the main difference of this work compared to existing explainable anomaly detection?
> >
> > - Following the suggestion of Reviewer yuBa to include discussions on recent and state-of-the-art explainers, I would like to expand upon this suggestion:  It would be better to compare the proposed explainable model with other explainable techniques, such as influence functions, Shapley value -based EAI, and post-hoc concept bottleneck models, etc. I understand that conducting these experiments during the rebuttal is impractical, but I wish the authors can consider this for future research. This is especially crucial and requisite when someone claims that he/she has provided a explainable model in any particular area or sub-area.
> >
> > - One of my prior questions appears to have received limited attention. It revolves around the observed instability and lack of robustness in the performance of all GLAD methods, characterized by extremely high standard deviations (Table 1). Furthermore, Table 2 indicates that OCGTL exhibits significantly greater stability compared to SIGNET. Could the authors delve into a more in-depth analysis to shed light on the underlying causes for these phenomena?
> >
> > - One minor suggestion: GLAD is used to represent "graph-level anomaly detection". However, in previous works they often use GAD instead. Better to stay consistent with previous works.

---

> > > ### Author Response · Authors · 2023-08-12
> > > **Response (1/2) to further questions raised by Reviewer 6qZe**
> > >
> > > We are grateful to Reviewer 6qZe for the valuable feedback. Below, you'll find our response addressing the raised questions. We hope that our response addresses your concerns.
> > >
> > > **Q5: Statement about “first attempt” contribution**
> > >
> > > **A5:** We deeply appreciate your valuable feedback and for sharing the recent survey paper in this research field. From the survey paper, we indeed identify two explainable papers related to graph-structured data: [214] for edge-level anomaly detection on dynamic graphs and [132] for node-level anomaly detection on network traffic data. To clarify this, we will use the term “graph-level anomaly detection” as our first-attempt contribution in the revised version, which will more our claim more accurate. It is still notable that we mainly focus on the “graph-level anomaly detection (GLAD)” problem which is highly distinct from the scenarios in the above papers. The core challenges in explainable GLAD problems, e.g., the lack of graph-level labels and the pattern diversity of subgraphs, are unique and ticklish. Since GLAD is a practical research problem, our contribution towards “self-interpretable GLAD” remains noteworthy and consequential.
> > >
> > > Regarding the applications of IB principle, the majority of studies focus on supervised graph/node/link classifications tasks where labels are available. In contrast, we apply this principle to an unsupervised anomaly detection scenario without the requirement of ground-truth labels. We believe that it is not a simple borrow-and-application task but requires well-crafted designs, such as multi-view MI maximization.
> > >
> > > To sum up, we will carefully modify our statement about the “first attempt” contribution to make the paper more precise. Appreciate again for the insightful comments.
> > >
> > > **Q6: Benchmark for comparison and metric**
> > >
> > > **A6:** We appreciate your concern regarding the benchmark and metric. For the benchmark for comparison, we found that GLocalKD[3] establishes the comparison on 16 datasets, including 10 datasets from TU datasets and 6 self-collected datasets (HSE, MMP, p53, etc.). Meanwhile, other representative studies (OCGIN [7] and OCGTL [8]) conduct experiments only on TU datasets. Such a conflict causes the difficulty of reproducing OCGIN and OCGTL on 6 new datasets with heavy grid-search to ensure fair comparison. In this case, we conduct our experiments on 10 commonly used TU datasets and 6 explainable datasets. We believe that our comparison is fair and reasonable.
> > >
> > > For the evaluation metric “AD-AUC”, we apologize for the unclear expression in the paper. Actually, “AD-AUC” is totally equivalent to “AUC” in previous papers [3,7,8]. Here we denote it as “AD-AUC” because we want to distinguish it from our explainability metrics (NX-AUC and EX-AUC) which are also called “AUC”. We will add more explanation for the metric in the revised paper and apology for the confusion.
> > >
> > > **Q7: Main difference compared to existing explainable anomaly detection**
> > >
> > > **A7:** We appreciate your feedback, and we will certainly update our related works section accordingly, referring to the recent survey papers and technical studies. As for the difference between our method and existing explainable anomaly detection (EAD), we would like to highlight the following two points:
> > >
> > > * Target task and explanation format. Existing EAD methods mainly focus on explaining anomaly detection results of image/tabular/time series data. To this end, they usually aim to learn explanations in the corresponding formats, such as pixel, feature value, and series. For the few EAD works for node/edge -level anomaly detection, their explanations mainly lie in node/edge features (mostly feature value). In contrast, we focus on the explainable GLAD problem, and our model aims to generate explanations in a subgraph format, including a group of nodes and corresponding edges. This is a more difficult task due to the discrete property of subgraphs and the complexity of graph-level patterns.
> > >
> > > * Technical solution. Our technical solution, i.e., multi-view subgraph information bottleneck, is novel and rarely seen in previous EAD methods before. Noting that existing EAD methods are mainly based on model gradient, approximation, reconstruction, etc. To the best of our knowledge, we are the pioneering study that applies multi-view learning and information bottleneck principle to EAD tasks. Although some components in our method are well crafted for graph-level EAD tasks, it is also promising to apply this learning framework to more EAD scenarios.

---

> > > > ### Author Response · Authors · 2023-08-12
> > > > **Response (2/2) to further questions raised by Reviewer 6qZe**
> > > >
> > > > **Q8: Combination with other explainable techniques**
> > > >
> > > > **A8:** We appreciate your valuable suggestion. We understand the importance of a thorough comparison. However, it's important to note that many of these explainers haven't been adapted for unsupervised graph-level anomaly detection. Consequently, applying these techniques to our specific learning scenarios is non-trivial. The discrete nature and non-Euclidean space of graph-structured data introduce challenges, particularly for certain EAI methods that produce continuous explanations. Furthermore, the absence of ground-truth labels in GLAD tasks exacerbates these complexities, especially given that most explainers are designed for supervised tasks. Moving forward, we will try our best to compare SIGNET with state-of-the-art methods.
> > > >
> > > > **Q9: Detailed analysis for experimental results**
> > > >
> > > > **A9:** We appreciate your continued engagement with our work and your insightful observation regarding the results in Table 1 and Table 2. In response to your feedback, we are committed to expanding the discussion in the revised version of our paper. A short explanation is provided as follows.
> > > >
> > > > For the high standard deviations in Table 1, we attributed it to the diversity of graph rationales and the lack of supervision signals. Note that the graph rationales (i.e., explanation subgraphs) in a dataset can be diverse. For example, the MUTAG dataset has two rationales “-NH2” and “NO2”, and BM-MS has cycle rationales in different sizes. In this case, without solid supervision signals to provide hints about such diverse patterns, it is difficult for an unpservised model like GLAD model to highlight them accurately. In some cases, the model may fail to capture the specific rationale patterns (e.g., the mutagenic molecules with multiple “-NH2” and “NO2”) while working well on other runs, leading to a high standard deviation. Such a phenomenon also inspires future works to develop more stable and robust explainable GLAD methods.
> > > >
> > > > In terms of the stability of anomaly detection performance, our results reveal that SIGNET demonstrates greater stability compared to OCGIN and GLocalKD, albeit slightly less stable than OCGTL. The key factor influencing stability is the number of GNN channels within a GLAD model. The presence of multiple channels diminishes the reliance on the individual quality of each channel, resulting in enhanced detection stability. The stability of OCGTL lies in its capacity to capture anomalous patterns by leveraging the consistency among multiple GNN channels, typically ranging from 4 to 6. SIGNET, the model with two GNN channels, has moderate stability compared to baselines. Conversely, other methods (OCGIN and GLocalKD) only incorporate a single learnable GNN channel, making their unstable performance, since the performance is susceptible to the quality of the only channel.
> > > > While the inclusion of multiple channels enhances stability, it also translates into increased computational demands and extended runtime for OCGTL. In forthcoming research, we may consider extending SIGNET to a multi-channel framework to improve its stability.
> > > >
> > > > **Q10: Abbreviation for “graph-level anomaly detection”**
> > > >
> > > > **A10:** We appreciate the reviewer for the valuable suggestion. We want to mention that GAD is commonly used as the abbreviation for “graph anomaly detection”, and in this paper, we use GLAD for “graph-level anomaly detection” to distinguish it from “graph anomaly detection” which mainly focuses on node-level anomaly detection.
> > > >
> > > > We do acknowledge the importance of maintaining consistency with prior works. As such, we will make the necessary adjustments to our abbreviations in the revised version to ensure coherence throughout the paper.

---

> > > > ### Comment · Reviewer_6qZe · 2023-08-13
> > > > **Thanks.**
> > > >
> > > > Thank you for your prompt response. I'd like to raise my score to 6.

---

> > > > > ### Author Response · Authors · 2023-08-17
> > > > > **Thanks for the raising**
> > > > >
> > > > > We are grateful to Reviewer 6qZe for suggesting this improvement! If you have any additional questions or concerns, please feel free to discuss them with us.

---

### Author Rebuttal · Authors · 2023-08-10

**General Response**

We sincerely thank all the reviewers for their valuable and insightful comments. We are glad that the reviewers find that the studied problem is novel and significant (Reviewer 6qZe, yuBa, and QzdR), the proposed method is novel and well-motivated (Reviewer 6qZe, yuBa, and vtQs), the theoretical analysis is sound (Reviewer yuBa), the empirical studies are adequate and reasonable (Reviewer vtQs), and the writing is smooth and has a good storyline (Reviewer yuBa, vtQs, and QzdR).

To the best of our efforts, we provided detailed responses to address the concerns raised by each reviewer in the following. Meanwhile, we carefully revised the paper according to the reviewers’ comments. We will incorporate all the feedback in the final version. ​​Specifically, the main revisions we made are as follows.

* We have added extra experiments to discuss the scalability on large-scale datasets and running efficiency of the proposed method (please see the Reply to Reviewrs 6qZe and QzdR for details).
* We have analyzed the time complexity of the proposed method (see the attached paragraph P1 below). The discussion shows that the complexity of the proposed method is comparable to mainstream GNN-based models.
* We have added more qualitative experiments, i.e., more visualization of explanation results, including results on real-world dataset (see the attached PDF).
* We have illustrated the implementation details of our baselines, i.e., the GLAD methods with post-hoc explainers (see the attached paragraph P2 below).
* We have added detailed explanations for our evaluation metrics, i.e., NX-AUC and EX-AUC (see the attached paragraph P3 below).
* We have highlighted the comparisons between our method (SIGNET) and a representative self-interpretable GNN, GSAT (see the attached paragraph P4 below).

**P1: Complexity Analysis of SIGNET.** Within this paragraph, we denote the average numbers of nodes and edges as $n$ and $m$ respectively, and denote the number of graphs and batch size as $N$ and $B$ respectively. At each training iteration, we first conduct DHT to obtain the dual hypergraph, which requires $\mathcal{O}(N(m+n))$. Then, the GNN-based extractor that calculates probability consumes $\mathcal{O}(NL_1md_1+NL_1nd_1^2 + Nnd_1d_f)$ complexity, where $L_1$ and $d_1$ are the layer number and latent dimension of the extractor, respectively. The bottleneck subgraph extraction for two views requires $\mathcal{O}(N(m+n))$ in total. For the GNN and HGNN encoders, their time complexities are $\mathcal{O}(NL_2md_2+NL_2nd_2^2 + Nnd_2d_f)$ and $\mathcal{O}(NL_2nd_2+NL_2md_2^2 + Nnd_2d_{f*})$ respectively, where  $L_2$ and $d_2$ denote their layer number and latent dimension. Finally, the Info-NCE loss requires $\mathcal{O}(NBd_2)$ complexity. To simplify the overall complexity, we denote the larger terms within $L_1$ and $L_2$ as $L$, and the larger terms between $d_1$ and $d_2$ as $d$. After ignoring the smaller terms, the overall complexity of SIGNET is $\mathcal{O}(NLd^2(m+n) + Nnd(d_f+d_{f*}) + NBd)$.

**P2: Implementation of GLAD methods with post-hoc explainers.** Given a GLAD model and post-hoc explainer, at first, we train the GLAD model independently on the training set. After sufficient training, the GLAD model is able to map each input graph into a scalar, i.e., its anomaly score. To address the uncertainty of the anomaly score boundaries, we apply a linear scaling function to map the scores into the [0,1] range and then use a sigmoid function to convert each score into a probability for binary classification. Subsequently, we integrate the post-hoc explainer with the probabilitized output of the GLAD model and optimize the explainer accordingly.

**P3: Evaluation metrics.** In this paper, we use “explanation Area Under the Curve (AUC)” to evaluate the explanation performance, following previous works [19,20]. We employ both node-level and edge-level explanation AUCs for comparison (NX-AUC and EX-AUC for short, respectively). Specifically, we tackle the explanation problem by framing it as a binary classification task for nodes and edges. We designate nodes and edges inside the explanation subgraph as positive instances and the rest as negative. The importance weights generated by the explanation methods serve as prediction scores. An effective explanation method should assign higher weights to nodes and edges within the ground truth subgraphs compared to those outside. To quantitatively evaluate the performance, we use the AUC as the metric for this binary classification problem. A higher AUC indicates better performance in providing meaningful explanations.

**P4: Comparison between GSAT and SIGNET**

Connections:
Both GSAT and SIGNET use information bottleneck principle as theoretical foundation of their explanation target that extracts the explanation subgraph. Both of them adopt neural networks to parameterize input graph and make the explanation differentiable, which is a common design among explainable GNNs.

Differences:
* Different targeted tasks: GSAT focuses on a graph classification where labels are available to train the interpretation module. Differently, SIGNET targets unsupervised GLAD, a more challenging task with unavailable labels during training.
* Different theoretical framework: GSAT is designed based on the original information bottleneck framework, tailored to its targeted supervised setting. In contrast, SIGNET is based on the multi-view subgraph information bottleneck (MSIB) framework derived in this paper, specifically designed for unsupervised GLAD.
* Different learning objectives: GSAT is trained using cross-entropy loss, a commonly used classification loss. In contrast, SIGNET is optimized using an Info-NCE loss, aiming to maximize the MI between each graph and its rational subgraph.
* Different graph view for graph learning: GSAT only considers the original view for graph learning, while SIGNET considers both the original and DHT views.

---

### Decision · Program_Chairs · 2023-09-21

**Decision:**

Accept (poster)

**Comment:**

The paper proposes the SIGNET approach for interpretable graph level anomaly detection. They first propose the multi-view subgraph information bottleneck (MSIB), which generates two views via dual hypergraph transform (DHT), then uses an information bottleneck approach to find the most informative subgraph as explanation.

Reviewers gave borderline to positive scores (6655). Strengths include the problem importance, well-designed method, and compelling empirical results. Issues raised include efficiency, novelty, and comparison to explainability baselines. In my view, the paper does introduce significant novel contributions, particular the focus on obtaining subgraphs for explaining graph anomalies; the method design utilizing information bottleneck to obtain subgraphs, and the use of dual hypergraph transform, which is an interesting and reasonable choice (as it is a deterministic and nontrivial transformation). Moreover, the case studies are interesting and show that the method indeed generates meaningful explanations, e.g. identifying the NH2 and NO2 subgraphs as explanations for the MUTAG dataset (matching ground truth), and cycles as explanations on some synthetic datasets.

For the remaining concerns, the rebuttal helped to address these satisfactorily, e.g. adding complexity analysis, experiments on larger datasets, additional baselines, and additional explanation results. Overall, reviewers and AC find the paper to be well-motivated by an important problem, novel, well-written, and with strong quantitative and qualitative empirical results. Hence, I recommend acceptance.

Authors are requested to note the suggested improvements / action items arising from discussion with the reviewers, such as adding the new empirical results to the final manuscript, and other clarifications to the paper arising from reviewer discussion (related work, hyperparameters, limitations, etc.)